# Purified regenerating retinal neurons reveal regulatory role of DNA methylation-mediated Na+/K+-ATPase in murine axon regeneration

Elias Rizk[1,2,4], Andy Madrid[1,4], Joyce Koueik[1], Dandan Sun [3], Krista Stewart[1], David Chen[1], Susan Luo[1], Felissa Hong [1], Ligia A. Papale[1], Nithya Hariharan[1], Reid S. Alisch [1,5✉] & Bermans J. Iskandar [1,5✉]

While embryonic mammalian central nervous system (CNS) axons readily grow and differentiate, only a minority of fully differentiated mature CNS neurons are able to regenerate injured axons, leading to stunted functional recovery after injury and disease. To delineate DNA methylation changes specifically associated with axon regeneration, we used a Fluorescent-Activated Cell Sorting (FACS)-based methodology in a rat optic nerve transection model to segregate the injured retinal ganglion cells (RGCs) into regenerating and non-regenerating cell populations. Whole-genome DNA methylation profiling of these purified neurons revealed genes and pathways linked to mammalian RGC regeneration. Moreover, whole-methylome sequencing of purified uninjured adult and embryonic RGCs identified embryonic molecular profiles reactivated after injury in mature neurons, and others that correlate specifically with embryonic or adult axon growth, but not both. The results highlight the contribution to both embryonic growth and adult axon regeneration of subunits encoding the Na$^+$/K$^+$-ATPase. In turn, both biochemical and genetic inhibition of the Na$^+$/K$^+$-ATPase pump significantly reduced RGC axon regeneration. These data provide critical molecular insights into mammalian CNS axon regeneration, pinpoint the Na$^+$/K$^+$-ATPase as a key regulator of regeneration of injured mature CNS axons, and suggest that successful regeneration requires, in part, reactivation of embryonic signals.

[1] Department of Neurological Surgery, University of Wisconsin-Madison School of Medicine and Public Health, Madison, WI 53792, USA. [2] Department of Neurological Surgery, Penn State Milton S. Hershey Medical Center, Hershey, PA 17033, USA. [3] Department of Neurology, University of Pittsburgh School of Medicine, Pittsburgh, PA 15213, USA. [4]These authors contributed equally: Elias Rizk, Andy Madrid. [5]These authors jointly supervised this work: Reid S. Alisch, Bermans J. Iskandar. ✉email: alisch@neurosurgery.wisc.edu; iskandar@neurosurgery.wisc.edu

The potential of CNS neurons to extend axons is greatest during embryonic development and follows a steady decline as development progresses[1], suggesting that embryonic molecular signals required for axon growth are not present in the mature neuron[2]. While peripheral nerves survive axotomy and retain the ability to grow axons as they mature[3,4], axonal injury in mature CNS tissue results in neuronal cell death[5,6]. However, mature CNS neurons can survive axotomy and a subset of these regrow long axons when the extracellular conditions are modified to provide a microenvironment permissive to axon growth[4,5,7–13], or when the expression of select neuronal developmental genes is enhanced[14–21]. These findings indicate that both extrinsic (glial) and intrinsic (neuronal) factors contribute to the capacity of neurons to survive and regenerate axons after injury. Intrinsic factors comprise a set of molecular signals in the neuron cell body that permit or suppress axon growth, referred to as the "growth state" of the neuron[22–25]. Conversely, extrinsic factors consist primarily of inhibitory molecules produced by the supportive CNS oligodendrocytes in the nascent white matter[26,27] and by astrocytes in post-injury "myelin scars."[27,28] In mature rodent optic CNS neurons (i.e., retinal ganglion cells [RGCs]), the inhibitory extrinsic factors can be diminished by attaching to the transected optic nerve a peripheral nerve graft. Because they are lined by Schwann cells instead of the oligodendrocytes that line CNS neurons, peripheral nerve grafts function as conduits that permit axon growth. Moreover, a growing body of research has found that satellite glial cells, a subpopulation of glial cells found in the peripheral nervous system, promote regenerative growth in peripheral neurons following injury[4,29,30], further underscoring the importance of glial cell recruitment and function in recovery in the mammalian nervous system. In the animal model reported here, approximately 10% of RGCs survive axotomy, and 10–15% of surviving neurons extend long axons into the graft[31,32]. Since the majority of RGCs still die or fail to regenerate axons in this permissive environment, the model underscores the importance of intrinsic neuronal regulatory mechanisms governing the expression of genes essential for CNS injury and regeneration[2,33–38].

DNA methylation (5-methylcytosine [5mC]) is an environmentally sensitive epigenetic modification that regulates gene expression. While 5mC is found in every mammalian tissue and cell type, its abundance and distribution are both tissue- and cell-type specific, resulting in the precise regulation of gene expression for critical biological processes including neuronal survival and synaptic plasticity[39–47]. CNS injury causes genome-wide alterations in DNA methylation levels[48–50], suggesting that modulation of this epigenetic mark may also contribute to CNS repair, including axon regeneration. However, studies reporting disruptions in DNA methylation levels following CNS injury and axon regeneration are confounded by data from bulk cell populations that are undergoing inflammation, apoptosis, and other biological responses to the neuronal injury, in addition to axon regeneration. In an effort to delineate DNA methylation changes specifically associated with axon regeneration, we developed a rodent method to segregate injured neurons based on their ability to regenerate optic nerve axons into an autologous peripheral nerve graft after optic nerve transection (Fig. 1)[51]. Data generated from this model provides evidence of a mechanistic role for DNA methylation in embryonic and mature RGC axon growth and regeneration, revealing that Na$^+$/K$^+$-ATPase subunits are key regulators of these processes.

## Results

**Selective FACS method identifies RGCs that regenerate axons after injury.** The study of molecular alterations associated with adult CNS axon regeneration following injury is limited by the challenge of separating signals of axon growth from inflammation, apoptosis, and other injury-associated molecular changes. We developed a fluorescence-activated cell sorting (FACS) protocol that segregates rat RGC neurons that are able to regenerate injured axons after optic nerve transection from those that are not. (Figs. 1 and 2a, b; Video 1). In this model, a sciatic nerve graft provides a permissive environment for transected optic axons to re-grow (regenerate). Axons that extend to the end of the graft are stained with a green-fluorescent tracer (Oregon green) that travels retrogradely to the RGC cell body (injured regenerated, or IR) (Figs. 1 and 2c; Video 3). It is well established that the fluorescent tracer does not diffuse down the graft; rather, when applied to the end of the graft, it is taken up by axons that have grown through the graft and transported retrogradely to the cell body in the retina[52–56]. Once the retina is removed and dissociated, all RGCs with or without regenerated axons are tagged with a Thy-1 neuronal antibody conjugated to a red fluorescent marker R-Phycoerythrin (Thy-1 PE) to separate RGCs away from non-neuronal retinal cells. Thus, the RGCs that are able to regenerate axons are labeled red and green (injured regenerated, or IR), whereas those that do not regenerate axons (injured non-regenerated, or INR) are labeled red only. Cells are next sorted based on relative size and granularity, then based on DAPI staining to discard dead cells, and lastly based on fluorescence (red vs. red/green, Fig. 2d, Video 3). Using this method, we selectively sorted Thy-1 PE-positive/Oregon Green-positive IR RGCs (389 ± 159 per retina, $N = 12$ retinas) and Thy-1 PE-positive/Oregon Green negative INR RGCs (117,567 ± 20,524 per retina, $N = 12$ retinas), confirming the literature that only a small subset of injured axons regenerate[31,32].

**DNA methylation levels segregate injured RGCs into regenerated and non-regenerated groups.** DNA was isolated from all FACS-sorted RGCs and treated with sodium bisulfite prior to whole-genome sequencing (Supplementary Data 1) to study DNA methylation abundance at >25 million CpG dinucleotides throughout the rat genome. An initial test for the role of DNA methylation in axon regeneration using unsupervised hierarchical clustering revealed that global DNA methylation levels distinguished IR and INR experimental groups from each other, suggesting that DNA methylation is an indicator of regenerative signals (Fig. 3a).

To identify genes and pathways associated with axon regeneration, the genome-wide DNA methylation data were subjected to a differential analysis that featured borrowing sequence read depth and methylation level information from neighboring CpGs using stringent criteria for significance (i.e., $P$ value <1e-4, methylation difference >10%, ≥3 consecutive significant CpGs). This approach identified 337 differentially methylated regions (DMRs) between IR and INR RGCs that were dispersed across all chromosomes of the rat genome, except on the Y chromosome and mitochondrial DNA (Fig. 3b; Dataset 1). Specifically, 151 DMRs contained hypermethylated sites (i.e., increases in DNA methylation levels in IR RGCs), while 186 DMRs exhibited hypomethylated sites (i.e., decreases in DNA methylation levels in IR RGCs). Similar to the hierarchical clustering of global DNA methylation levels used to stratify IR and INR RGCs, data from only the CpGs located in DMRs successfully segregated IR from INR RGCs (Fig. 3c). The distribution of DMRs across standard genomic structures revealed that the majority (~57%) were >10 kilobases away from any gene (i.e., in intergenic regions), and the next largest subset resided in intronic regions (~28%; Fig. 3d). Since these regions often harbor gene regulatory elements (i.e., enhancers), these

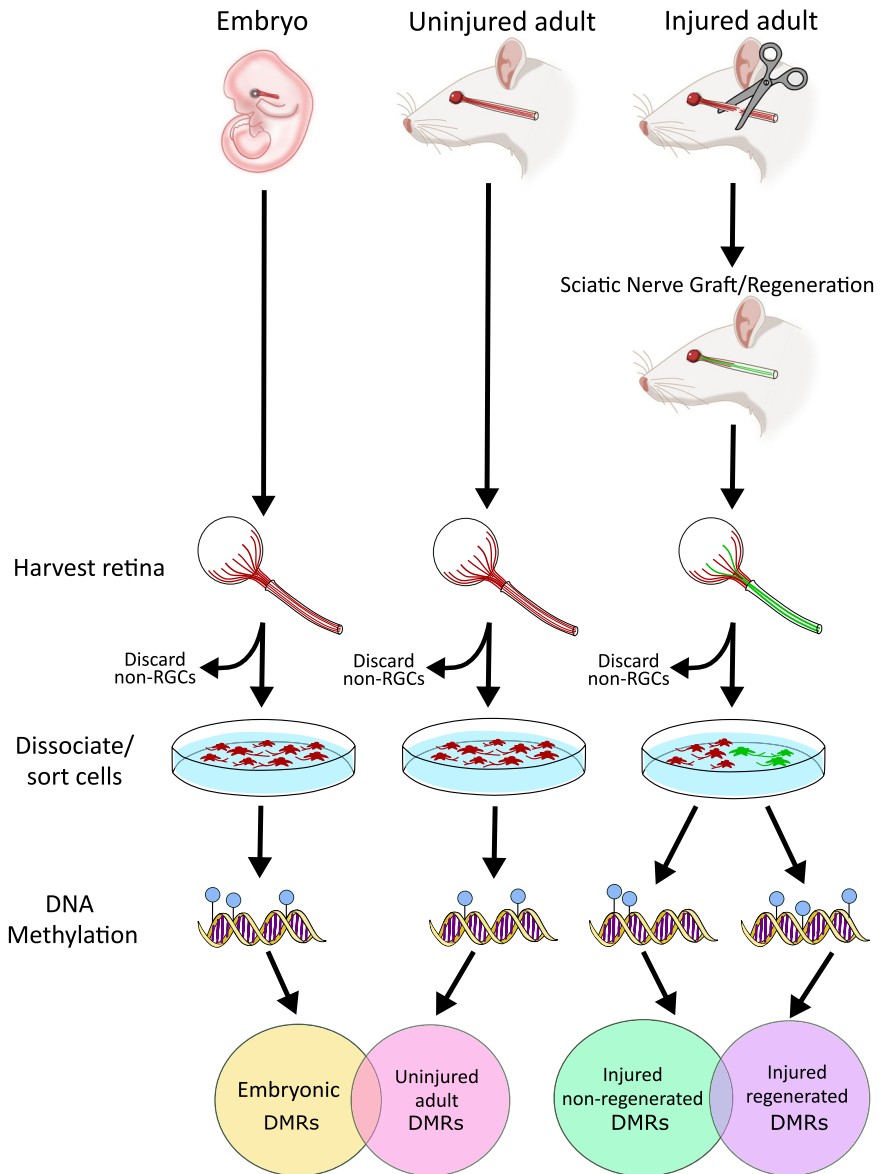

**Fig. 1 Overview of the experimental design.** A visual flowchart depicts the overall design of the experiments reported in this article. The first column depicts experiments using uninjured embryo (UE) retinal ganglion cells (RGCs). Embryonic retinas were harvested and dissociated, then FACS-sorted and sorted into RGCs (stained red) and non-RGCs (unstained). The non-RGCs were discarded. DNA was extracted from sorted RGCs and whole-methylome sequencing was performed. The second column depicts experiments using uninjured adult (UA) RGCs. Adult retinas were harvested and dissociated, then FACS-sorted into RGCs (stained red) and non-RGCs (unstained). The non-RGCs were discarded. DNA was extracted from sorted RGCs and whole-methylome sequencing was performed. The third column depicts experiments using injured adult RGCs. See Fig. 2 for details of the surgery, nerve grafting, and fluorescent staining. The retinas were harvested, dissociated, and FACS-sorted, generating Injured/regenerated (IR; green) and injured/non-regenerated (INR; red) RGCs. DNA was extracted from sorted RGCs and whole-methylome sequencing was performed. Venn diagrams depict DMR overlaps between uninjured adult and embryonic RGCs (left Venn), and between injured regenerated and non-regenerated RGCs.

findings suggest that axon regeneration-associated DMRs may play a key role in regulating gene expression of target genes by regulating transcription factor binding to enhancer elements. Thus, the genomic sequences comprising these DMRs were subjected to a motif enrichment analysis to identify transcription factor binding sites that might be affected by altered DNA methylation abundances. The hypermethylated and hypomethylated regions were significantly enriched for several transcription factor motifs including DLX5 and ELF5, respectively (Fig. 3e, f; Supplementary Data 1), both of which serve as regulators of regeneration in other tissues[57,58], along with processes specifically related to the observed regenerative phenotypes of RGCs, including stem cell differentiation and self-renewal[59,60]. These

data support a mechanism for DNA methylation that influences the binding affinity of regeneration-associated transcription factors and alters the expression levels of target genes.

**Identification of differentially methylated genes associated with RGC regeneration.** Annotation of the DMRs to the nearest gene revealed 59 and 87 genes with increased and decreased methylation abundances in IR (vs. INR) RGCs, respectively. Gene ontology identified biological processes displaying enriched sets of DMR-associated genes. The top ontological terms derived from hypermethylated genes were associated with the development of the eye and the visual system, such as retina homeostasis and

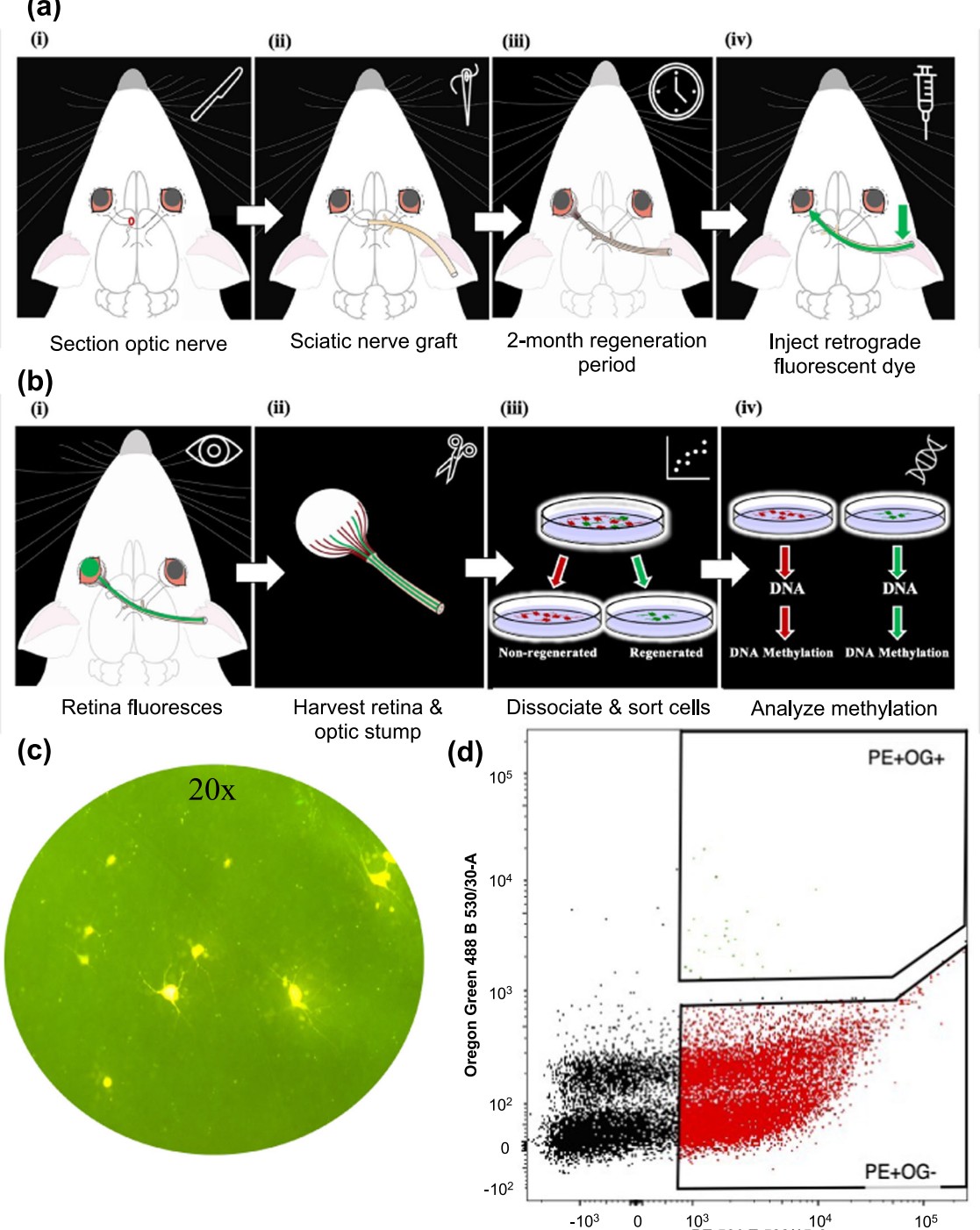

**Fig. 2 Schematic of the experimental paradigm and the neuronal isolation protocol. a** (i) Six-week-old rats were anesthetized, and the optic nerve of the left eye was sectioned. (ii) A portion of the sciatic nerve was grafted to the optic stump. (iii) An eight-week period was allowed for injured RGCs to regenerate axons to reach the sciatic nerve graft. (iv) In animals to be used for phenotyping (quantitation of RGC regeneration), the free end of the graft was backfilled with the retrograde tracer Fluorogold (see C). In animals used for cell sorting, the free end of the graft was backfilled with the retrograde tracer Oregon green. **b** (i) Fluorescent retina. (ii) The retina and optic stump destined for cell sorting were harvested. (iii) RGCs were dissociated and stained with the neuronal Thy-1 marker conjugated to fluorescent R- Phycoerythrin (Thy-1 R-PE). Dead cells were removed using DAPI, and the live cells were sorted with FACS. Red arrow—sciatic nerve; blue arrow—green-fluorescent tracer (Oregon green) application site. (iv) Following cell sorting and purification of regenerated and non-regenerated RGC populations, DNA was extracted from cells and used to investigate DNA methylation profiles. **c** The retina destined for phenotyping was harvested, flat-mounted, and visualized under a fluorescence microscope. A representative image (×20) of the retina with fluorescent RGCs (Fluorogold-positive) indicating that they regenerated axons to the end of the graft. (d) A scatterplot shows the distribution and abundance of dead cells (black dots), live injured/regenerated (IR) RGCs (green dots; Thy-1 R-PE[+]; OG[+]), and live injured/non-regenerated (INR) RGCs (red; Thy-1 R-PE[+]; OG[−]) across the red (x-axis) and green (y axis) fluorescent intensities.

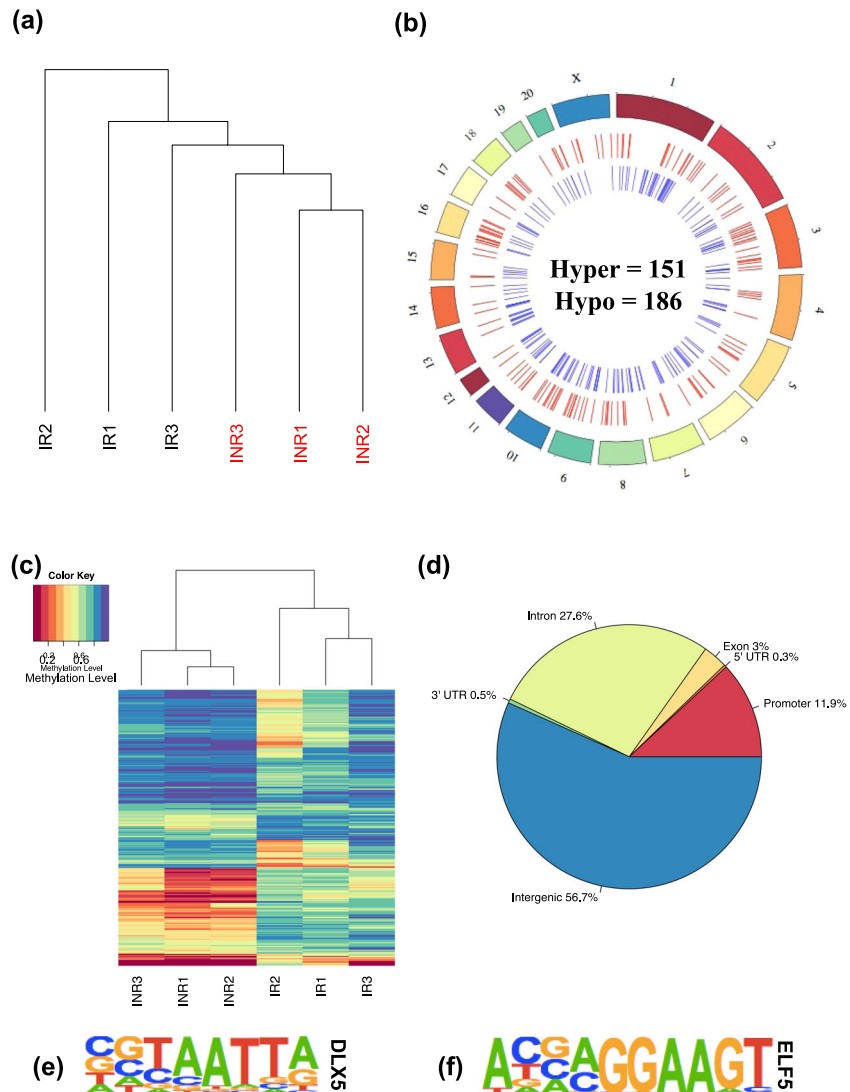

**Fig. 3 DNA methylation levels are linked to regeneration following injury. a** Unsupervised hierarchical clustering using global levels of DNA methylation levels distinguish injured/regenerated (IR) ($N = 12$ retinas) (black) from injured/non-regenerated (INR) ($N = 12$ retinas) RGCs (red). **b** A circos plot depicts the genome-wide relative location of regeneration-associated hyper DMRs ($N = 151$; red; middle circle) and hypo DMRs ($N = 186$; blue; inner circle) on each chromosome (outer circle) across the genome. **c** Unsupervised hierarchical clustering using the DNA methylation data from only the CpGs located in the 337 DMRs comparing IR against INR RGCs. Corresponding heatmap shows the level of methylation at each CpG from each RGC group. Low levels of DNA methylation (red) and high levels of DNA methylation (purple) are depicted. **d** A pie chart shows the proportion of DMR locations relative to standard genic structures. **e, f** Weblogos display the motif of a representative transcription factor significantly enriched in hypermethylated (DLX5) and hypomethylated (ELF5) regions.

photoreceptor cell maintenance (Fig. 4a). Top ontological terms stemming from hypomethylated genes were predominantly associated with dendrite development and morphogenesis (Fig. 4b).

To validate this DMR identification approach, we performed a more granular analysis (i.e., FDR $P$ value <0.01, methylation difference >20%) and examined differentially methylated loci (DMLs) instead of regions (DMRs). We observed that ~60% of the 65,204 DMLs in the IR and INR comparison (Supplementary Data 1) reside >10 kilobases away from an annotated gene in the rat genome (Supp. Fig. 1b), and ontology of the genes harboring DMLs revealed that the top terms were related to eye developmental processes (Supp. Fig. 1c, d). While genes responsible for *de novo* DNA methylation, namely *Dnmt3a*, *Dnmt3b*, displayed differential methylation, genes involved in DNA demethylation processes (i.e., *Tet1*, *Tet2*, and *Tet3*) were not differentially methylated in regenerated RGCs. These data confirm that regeneration-related differential methylation is

agnostic to the analytical approach, as similar distributions, genes, and pathways were identified from both DMLs and DMRs.

Next, we determined the expression level of genes known to function in axon regeneration and found that several hypomethylated genes exhibited induced expression levels in IR compared to INR RGCs, including *Cd38*, *Rassf4*, *Nav3*, and *Il4r* (Supplementary Data 1). Conversely, several hypermethylated genes had reduced expression levels in IR compared to INR RGCs, such as *Negr1* and *Nefl*. We also found an example of a hypomethylated gene with reduced expression (*Arhgap20*; Supplementary Data 1). These data support a functional role for DNA methylation in regulating gene expression that governs axon regeneration in mature RGCs.

**DNA methylation levels in adult injured RGCs mirror those of uninjured embryonic RGCs.** To determine whether the altered

## (a) Hypermethylated DMR-associated Genes

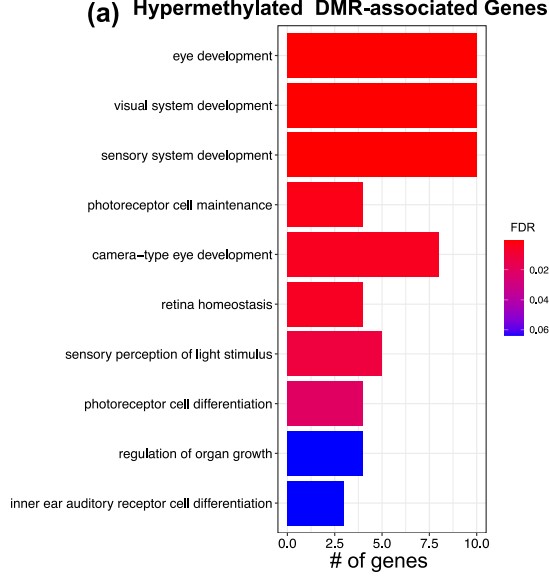

## (b) Hypomethylated DMR-associated Genes

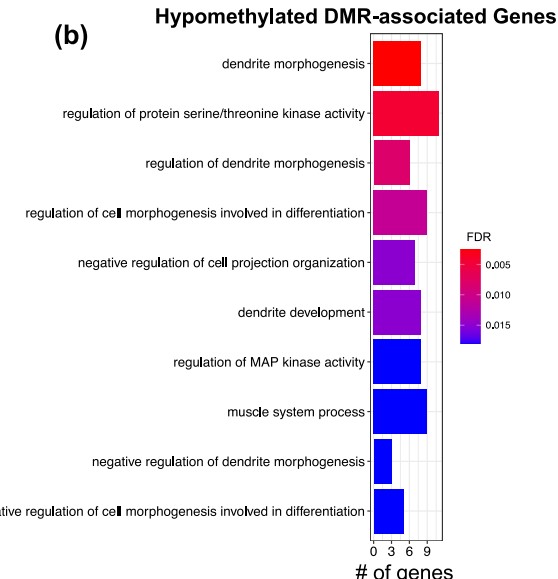

**Fig. 4 Pathways and potential regulatory features of regeneration-related DMRs. a, b** barplots show the top 10 ontological terms (y axis) linked to genes that contain hypermethylated (Hyper) and hypomethylated (Hypo) regeneration-associated DMRs, respectively. Bar color depicts the FDR P value for each term. Bar size (x axis) represents the number of DMR-associated genes identified for each ontological term.

DNA methylation levels in adult injured RGCs represent the reactivation of embryonic epigenetic signals activated by injury, we first aimed to determine the differences in DNA methylation levels between uninjured adult and embryonic RGCs. To that end, we isolated uninjured embryonic (UE) and uninjured adult (UA) RGCs that were FACS-sorted from non-RGCs, purifying $150,000 \pm 86,603$ embryonic cells (UE) per retina ($N = 12$ retinas) and $166,667 \pm 96,225$ adult cells (UA) per retina ($N = 12$ retinas) (Fig. 1; Video 2). The DNA from these RGCs was treated with sodium bisulfite prior to whole-genome sequencing to provide DNA methylation levels at more than 25 million CpG dinucleotides throughout the RGC genomes (Supplementary Data 1). A genome-wide differential analysis of the UE and UA DNA methylation data revealed 753 and 1,229 unique genes containing hyper- and hypomethylated DMRs (relative to UE samples), respectively (Fig. 5a, b; Dataset 1). Gene ontology revealed

significant enrichments of processes involved in visual system function including retina homeostasis, visual perception, eye morphogenesis, and developmental processes including axon guidance, forebrain development, connective tissue development, and visual system development (Fig. 5c, d). These data present evidence that genes with greater DNA methylation abundance in embryonic RGCs are linked to processes involved in eye development, while genes displaying lower DNA methylation abundance in embryonic RGCs are associated with general system-level developmental processes.

Subsequently, to determine the methylomic intersection between axon growth in the developing embryonic CNS and axon regeneration in the injured mature CNS, we compared DMR-associated genes linked to adult axon regeneration (i.e., IR compared to INR) to DMR-associated genes linked to embryonic axon growth (i.e., UE compared to UA; Fig. 6; top panel). This comparison revealed 48 genes with shared adult and embryonic axon growth potentials. Of these, the $Na^+/K^+$-ATPase is the family of genes most linked to axon regeneration mechanisms in the literature[61–67]. Specifically, the $Na^+/K^+$-ATPase subunit gene *Atp1b2* displayed hypermethylation in both growing UE RGCs and regenerating adult RGCs, implying that the $Na^+/K^+$-ATPase may reactivate its embryonic DNA methylation levels following injury to facilitate axon regeneration.

*The expression of $Na^+/K^+$-ATPases after injury is higher in regenerated RGCs compared to non-regenerated RGCs.* Numerous DMR/DML-associated genes identified in our analyses encode subunits of the $Na^+/K^+$-ATPases, including *Atp1a1*, *Atp1a2*, *Atp1a4*, *Atp1b1*, *Atp1b2*, *Atp1b3*, and *Atp1b4*. All but one subunit (*Atp1b4*) exhibited hypomethylation in IR compared to INR. Examination of the DNA sequences immediately flanking ($+/-125$bp) the DMLs associated with the differentially methylated $Na^+/K^+$-ATPase subunits revealed that 11/14 flanks contain a putative binding motif related to ELF5 (GGA[AT]), while one flank fosters a DLX5 binding (TAATTA) motif. These data suggest that regeneration-associated differential methylation may alter the expression $Na^+/K^+$-ATPase subunits by altering the binding affinity of transcription factors, warranting further investigation of $Na^+/K^+$-ATPase subunits expression levels during axon regeneration following injury.

We then measured the mRNA and protein levels of the $Na^+/K^+$-ATPase's main subunits in IR and INR RGCs. Using quantitative RT-PCR, the $\alpha_1$ and $\beta_1$ subunits exhibited a 2.8-fold and 8.7-fold higher mRNA level in IR compared to INR, respectively (Fig. 7ai, aii). Furthermore, immunoblotting using monoclonal antibodies against subunits of the $Na^+/K^+$-ATPase showed a 2.8-fold increase in the α1 subunit, 1.3-fold increase in the 60-kDa β subunit, and 1.7-fold increase in the 46-kDa β subunit in IR compared to INR (Fig. 7b; Supplementary Data 1). These results support a role for differential expression of $Na^+/K^+$-ATPase in axon regeneration, which may be mediated by modulations in DNA methylation abundance affecting transcription factor binding.

**Inhibition of $Na^+/K^+$-ATPase activity reduces post-injury RGC axon regeneration.** Finding that several $Na^+/K^+$-ATPase subunits exhibit correlated differential DNA methylation, RNA, and protein levels in regenerating RGCs led us to test whether $Na^+/K^+$-ATPase activity is essential for axon regeneration. Digoxin is an inhibitor of the $Na^+/K^+$-ATPase pump. Adult rats that were treated with intraperitoneal injection of digoxin ($N = 8$) or DDI control ($N = 7$) were subjected to the optic nerve transection method, and the number of RGC neurons that regenerated axons into a peripheral nerve graft was counted. There was a

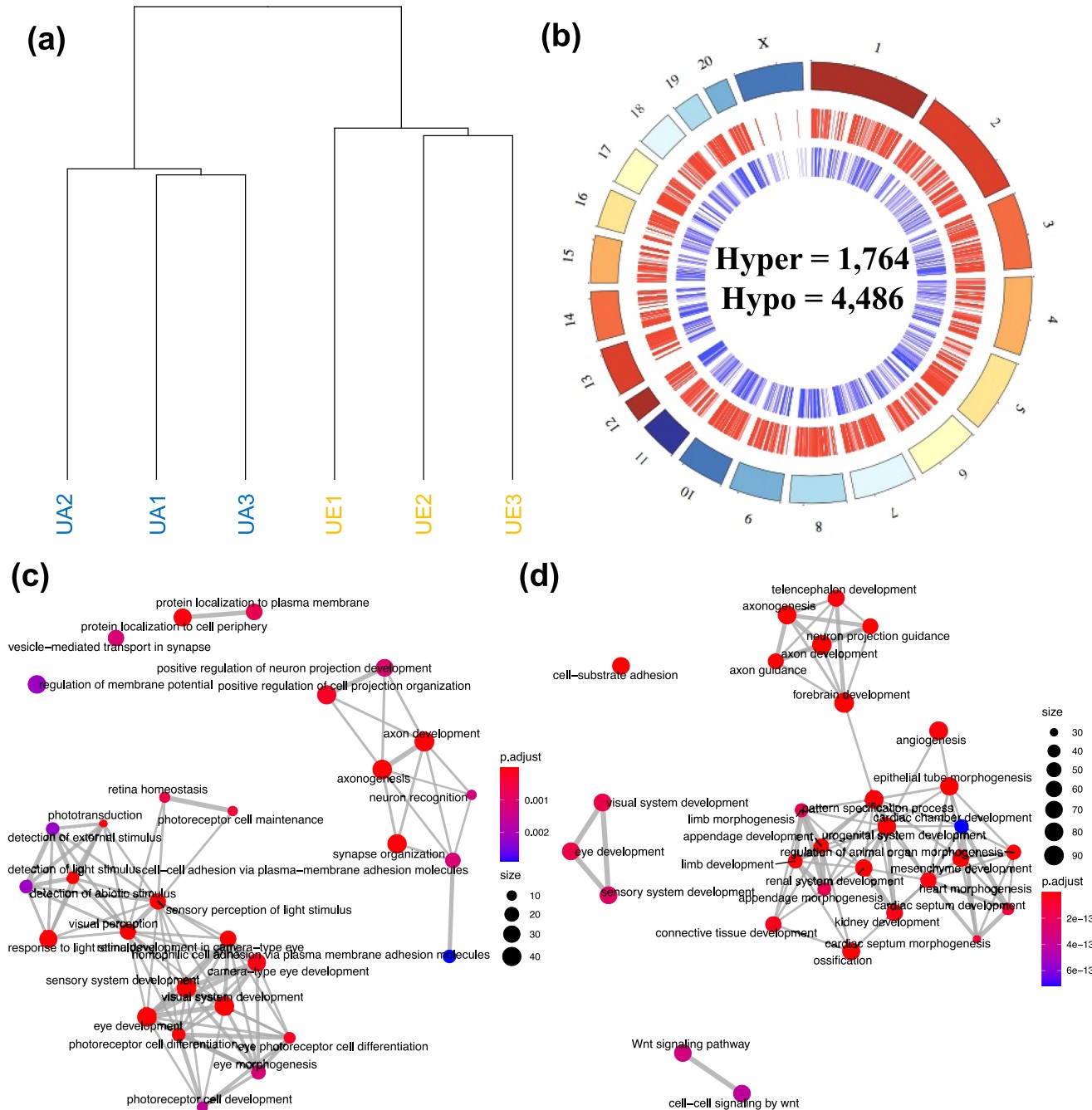

**Fig. 5 Differential methylation levels distinguish embryonic (UE) from adult (UA) uninjured RGCs. a** Unsupervised hierarchical clustering using global levels of DNA methylation distinguish uninjured adult (UA) (blue) ($N = 12$ retinas) from uninjured embryonic (UE) (orange) ($N = 12$ retinas) samples. **b** A circos plot depicts the genome-wide relative location of hypo DMRs ($N = 4486$; red; middle circle) and hyper DMRs ($N = 1764$; blue; inner circle) in association with the chromosomes (outer circle) across the genome. **c**, **d** Enrichment map plots show the interconnectivity of top ontological terms derived from hypermethylated and hypomethylated genes, respectively, that differentiate UA from UE samples. Color of dots represents the adjusted *P* value for each of the displayed ontological terms. Size of dots represents the relative number of genes associated with each displayed ontological term.

significant reduction in the number of regenerated axons (*P* value <0.05; Digoxin: $605 \pm 190$ and DDI: $895 \pm 163$ per retina [mean ± SE]) following Na⁺/K⁺-ATPase pump inhibition (Fig. 7c).

*Heterozygous knockout of Na⁺/K⁺-ATPase subunits reduces post-injury RGC axon regeneration.* As digoxin is a non-specific inhibitor of the Na⁺/K⁺-ATPase pump, we sought to confirm the importance of Na⁺/K⁺-ATPase in RGC axon regeneration

using a genetic technique. We subjected two independent Na⁺/K⁺-ATPase α-subunit heterozygous mouse models (i.e., *Atp1a1⁺/⁻* and *Atp1a2⁺/⁻*) to the optic nerve transection method as previously described. Both heterozygous mouse models showed significantly less regenerative capacity than wild-type controls. Specifically, the α1 heterozygous knockouts showed $165 \pm 38.5$ ($N = 6$, mean ± SE) regenerated RGCs per retina, compared to $403 \pm 78$ ($N = 10$) in wild-type controls, a 40% reduction

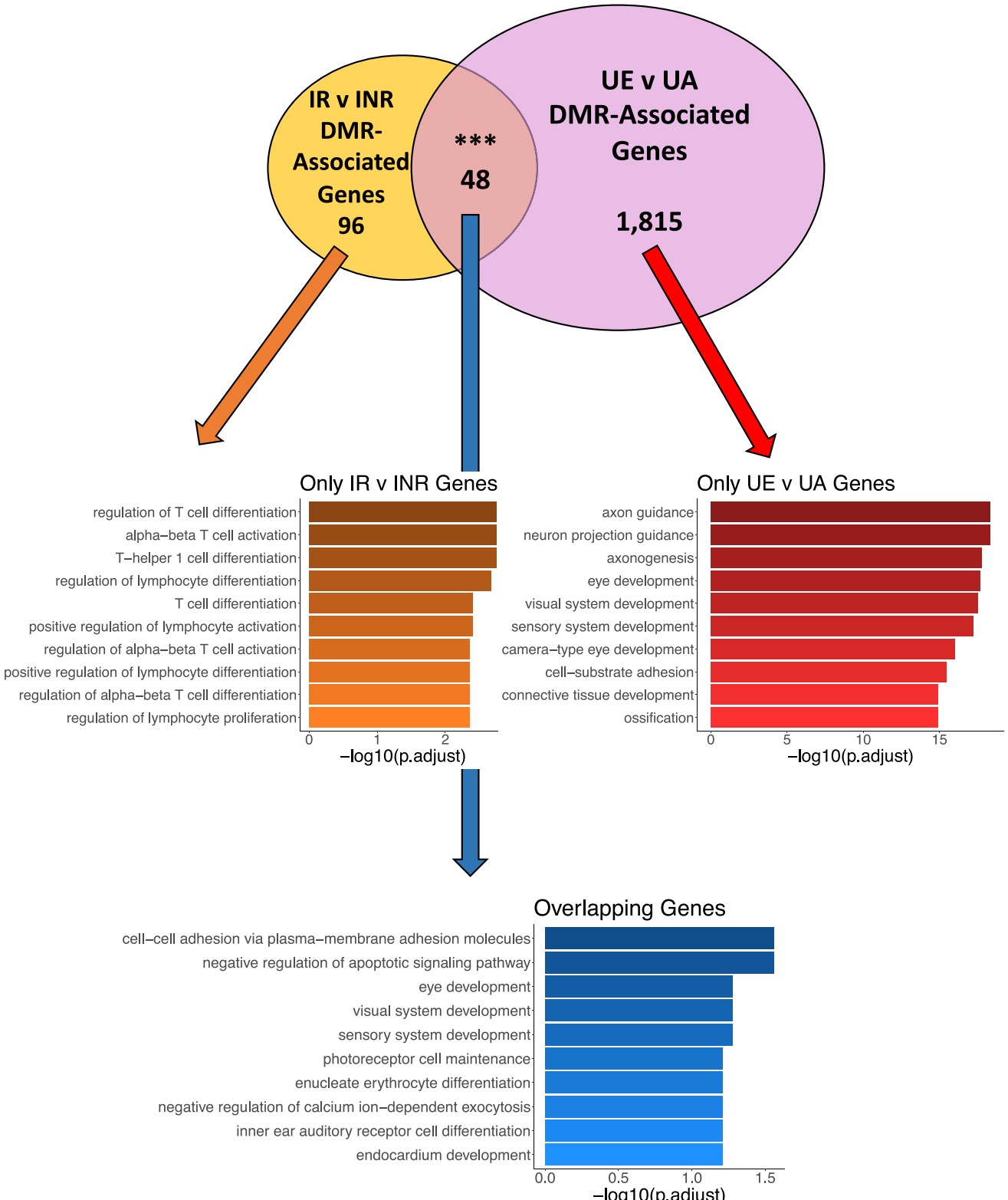

**Fig. 6 Distinct processes are affected in embryonic and regenerated adult RGCs.** A Venn diagram shows the overlap ($N = 48$) between genes containing DMRs in the injured/regenerated (IR) and injured/non-regenerated (INR) samples compared to the uninjured adult (UA) and uninjured embryonic (UE) samples. Dot plots for each unique set of genes are depicted below the Venn diagram. The color of the dots represents the adjusted $P$ value from the enrichment analysis. The size of the dots represents the number of genes associated with each ontological term ($y$ axis). The $x$ axis depicts the percent of DMR-associated genes contained among all the genes associated with each ontological term.

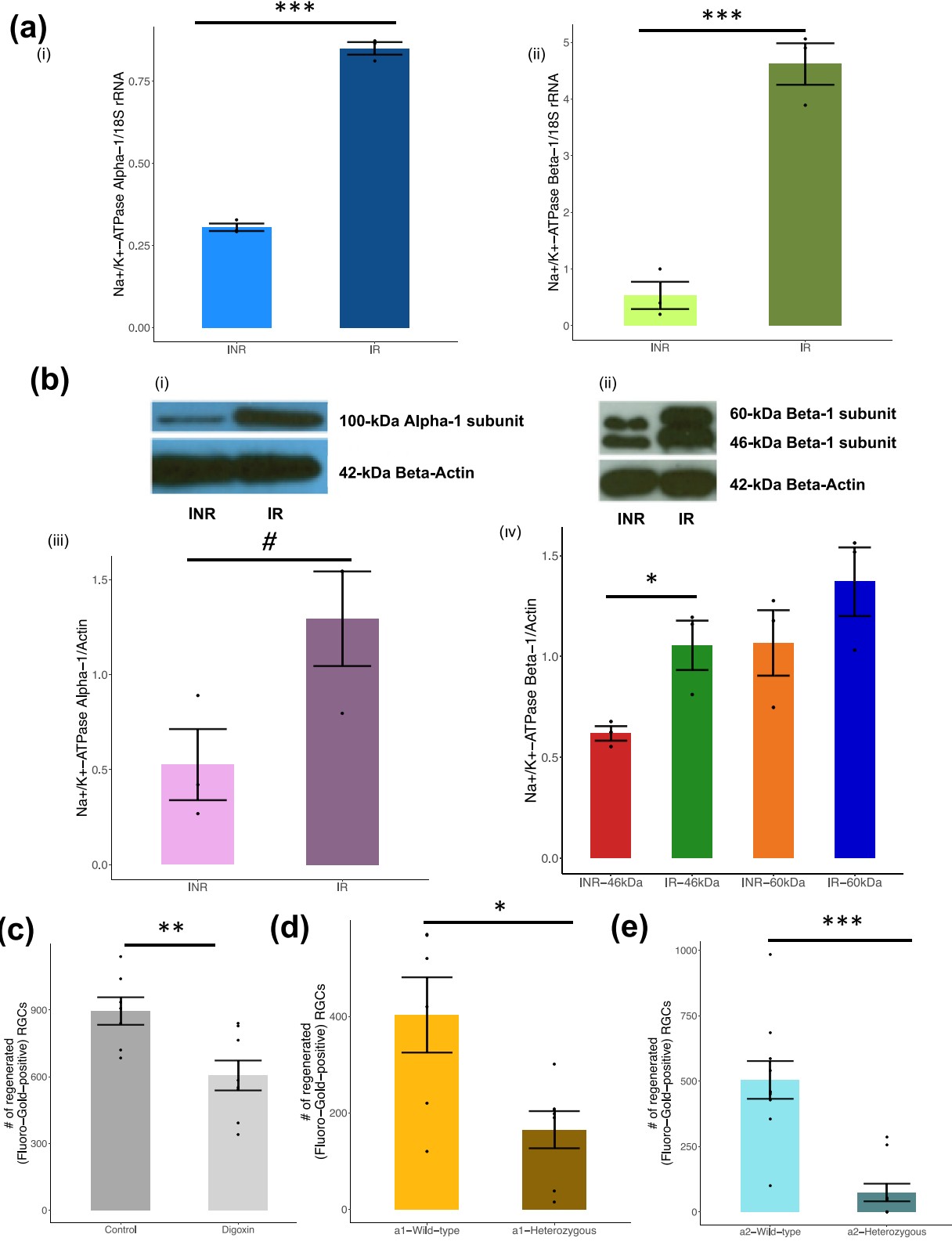

(Fig. 7d), and the α2 heterozygous knockouts had $74 \pm 33.7$ ($N = 7$) regenerated RGCs compared to $504 \pm 72$ ($N = 10$) in wild-type controls, an 85% reduction (Fig. 7e).

## Discussion

**Separating molecular signals related to axon regeneration from those related to neuronal injury**. Distinguishing the molecular

signals involved in regeneration of injured CNS axons from those related to injury mechanisms is challenging. We developed a method to separate neurons based on their ability to regenerate axons into an autologous peripheral nerve graft attached to a transected rat optic nerve. Until now, studies profiling regeneration signals in the optic nerve had relied on a comparison between injured (usually optic nerve crush or transection) and

**Fig. 7 Regenerated RGCs overexpress Na⁺/K⁺-ATPase and show increased pump activity compared to non-regenerated RGCs after injury. a** RT-PCR analyses of injured/non-regenerated (INR) and injured/regenerated (IR) RGCs ($N = 3$ in each group). Results show Na⁺/K⁺-ATPase-α (i) and β (ii) mRNA expression levels relative to 18 S RNA expression levels (y-axis) in INR and IR RGCs (x axis). **b** Immunoblot analyses of INR and IR RGCs ($N = 3$ in each group). INR and IR RGCs were lysed and immunoblots of the cell lysates were probed using monoclonal antibodies raised against the α-subunit (i; 100-kDa) and β-subunit (ii; 46-, 60-kDa) of Na⁺/K⁺-ATPase, and anti-β-actin (42-kDa, panel below each blot). The protein expression was normalized against β-actin. Quantification of the immunoblots for the α-subunit (iii) and β-subunit (iv) of Na⁺/K⁺-ATPase. Full blot images are provided in Supplementary Data 1. **c** The number of regenerated axons (y axis) amongst the control ($N = 7$; 895 ± 163) and digoxin-treated ($N = 8$; 605 ± 190) animals (x axis) are depicted. A significant reduction in the number of regenerating cells was found following digoxin-mediated inhibition of the Na⁺/K⁺-ATPase. Values shown are mean ± SEM and all asterisks indicate a P value <0.05 from a Student's t test. **d, e** The number of regenerated RGC axons in α1 (**c**; $N = 6$) and α2 (**d**; $N = 10$) heterozygous knockout mice (y axis) was compared to those in α1 (**c**; $N = 7$) and α2 (**d**; $N = 10$) wild-type control mice (x axis). Values shown are mean ± SEM and all asterisks indicate a P value <0.05 from a Student's t test. Figure 1c and Video 1 show a representative image (×20) of retina flat mount with the fluorescent (Fluorogold-positive) RGCs that regenerated axons to the end of the graft. The fluorescent cells are counted under fluorescent microscopy to generate the data in **c–e**.

uninjured RGCs, without specifically segregating injured neurons that regenerated from those that have not. A similar FACS approach was used to separate retinal cells based on cell type (e.g., RGC vs. non-RGC) and injury state[68,69], while others separated the neurons based on their stage of development[70]. Molecular signatures of CNS tissues that are known to regenerate after injury (frog eye and tadpole hindbrain) were compared to others that are incapable of such regeneration (e.g., frog hindbrain)[64]. And more recently, RGC sorting was done by profiling cells based on genome-wide loss-of-function screen for factors limiting axon regeneration after optic nerve crush[71]. This complex method identified genes that inhibit, but not those that enhance, axon regeneration. This and other single-cell RNA-sequencing technologies and transcription factor screens have yet to provide researchers with the ability to profile cells based on their proven ability to regenerate axons in vivo, underscoring both the novelty and relative simplicity of our method[71–73]. This approach confirms previous literature that only 10% of RGCs survive axotomy, and 10–15% of those exhibit regenerative properties[31,32]. Notably, fluorescent tracers, such as those applied to the end of the graft (i.e., Oregon green and Fluorogold), do not diffuse through the peripheral nerve graft. Instead, axons that grow through the graft (i.e., regenerate) and reach its free end are selectively labeled by transporting the fluorescent tracer retrogradely[52–56]. Whole-genome bisulfite sequencing of the segregated neuronal groups revealed epitypes linked to adult RGC axon regeneration. Ontology of the regeneration-associated differentially methylated genes showed strong biological relevance to developmental and axon regeneration functions. This is the first study to purify regenerated from non-regenerated neurons following injury, selectively sequestering regenerated cells from non-regenerating cells to molecularly profile associations with regeneration that are not confounded by other injury-related signals. The analyses revealed known (i.e., *Atp1a1*) and novel (i.e., *Sag*) genes that are molecularly implicated in mammalian CNS regeneration.

**Unique genes and processes are involved in adult RGC axon regeneration.** Few studies have thoroughly investigated the role of DNA methylation in axon regeneration, particularly in relation to neuronal growth competence[19,35,38,48,74]. Recent studies have focused on transcriptional alterations associated with neuronal injury[33,75–78]. An optic nerve crush model revealed 46 subclasses of RGCs based on transcriptional and morphological changes that precede degeneration, with a subgroup of genes showing particular resilience or susceptibility to injury[33]. Moreover, loss- and gain-of-function assays revealed an association between cell-type-specific responses to injury, neuronal survival, and axon regeneration, including genes that exhibited differential methylation in our study (e.g., *Foxp2*, *Sdk2*, and *Elf4*)[33].

In this study, we show that both hypermethylated and hypomethylated genes are functionally related to tissue regeneration processes, including eye development (e.g., *Cacna1c*, *Gnb1*, *Prdm1*), retina homeostasis (e.g., *Nxnl1*, *Pde6a*, *Cdh23*), and organ growth regulation (e.g., *Notch1*, *Nrg1*, *Hlx*) among the hypermethylated genes[79–81], and dendrite morphogenesis (e.g., *Adrb2*, *Dtnbp1*, *Ptprc*), MAP kinase activity (e.g., *Paqr3*, *Rgs2*, *Tnik*), and protein serine/threonine kinase activity (e.g., *Prkag2*, *Uvrag*, *Mdfic*) among the hypomethylated genes[82–84]. These data suggest that DNA methylation signatures may serve as an indicator/regulator of RGC axon regeneration potential. Moreover, we identified differential methylation and gene expression changes on several well-known regeneration-associated genes, including *Cd38*, *Rassf4*, *Nav3*, *Il4r*, *Negr1*, *Arhgap20*, and *Nefl*. The observation that DNA methylation alterations in these regeneration-associated genes were coupled with changes in gene expression provides evidence of a functional role for DNA methylation in axon regeneration.

DNA methylation plays a critical role in regulating transcription by recruiting or hindering protein (e.g., transcription factors) binding to DNA, and/or by altering chromatin structure[85]. Unique sets of significantly enriched transcription factor binding site motifs were identified among the hyper- and hypomethylated DMRs, including hypermethylated sites putatively bound by distal-less homeobox 5 (DLX5), which encodes a homeobox transcription factor with known functions in bone development and healing following fracture[86,87], regulation of stem cell pluripotency[59,88], behavior, metabolism, and aging processes[89], development of neural crest cells[90] and neural tube closure[58]. Significant enrichments in the hypomethylated regions included E74-like ETS transcription factor 5 (ELF5), which has known roles in regulating terminal differentiation of keratinocytes, and trophoblast stem cell self-renewal and differentiation[91,92]. These examples suggest that RGCs undergoing axon regeneration are entering a "growth-competent" state via a DNA methylation-mediated shifting in transcription factor binding. Future studies of chromatin structure would clarify the contribution of DNA methylation-mediated long-range interactions between enhancers, transcription factors, and gene promoters that promote/inhibit CNS regeneration.

Molecular profiles examined in this study only interrogated one timepoint, which was two months following optic nerve transection and transplantation of a peripheral nerve graft onto the optic stump. The two-month timepoint was chosen because this is when the number of regenerating axons reaches a maximum before a plateau. Of course, it is likely that over the course of this two-month period, the molecular architecture of regenerating axons changes, therefore examining only one cross-sectional point may not fully capture the extent of molecular alterations contributing to the continuous process of axon

regeneration. However, it is hypothesized that molecular and cellular regenerative mechanisms remain activated until the regenerating axons reach their target[93]. In our animal model, the end of the graft lies freely in the subcutaneous space with no target attachment. Therefore, while studying shorter and longer timepoints is of interest, investigating the molecular changes at two months post-injury should encompass many of the signals regulating and differentiating regenerated from non-regenerated axons.

**Role of embryonic molecular signals in axon regeneration of injured adult RGCs.** While we observed that the methylation patterns of a subset of genes overlap between adult RGC regeneration (IR v INR) and embryonic growth (UE v UA) ($N = 48$), the vast majority of genes that exhibit differential methylation are unique to either adult ($N = 96$) or embryonic ($N = 1815$) RGC growth. Previous research discerned extensive differences between the transcriptomes of human embryonic stem cells (hESCs) and those of hESC-derived RGCs[94], with strong similarities between hESC-derived RGCs and adult human RGCs. Moreover, open chromatin regions enriched in embryonic cells were found to harbor binding motifs for transcription factors with potential roles in axon growth (e.g., CREB) of postnatal RGCs. In turn, overexpression of CREB fused to the VP64 transactivation domain in RGCs was found to promote axon regeneration following optic nerve injury[95], confirming that reactivation of embryonic signals may enhance axon regeneration in adult neurons. Recent research using single-cell RNA-sequencing technologies has examined the temporal changes in gene expression across embryonic and postnatal development[96]. Using 3 embryonic (E13, 14, and 16) and 3 postnatal (P0, 5, and 56), researchers identified 6 sets of genes with altered expression across embryonic and postnatal development, shifting from gene network programs primarily associated with axon guidance (early embryonic module) to synaptic refinement processes (late postnatal module). These results highlight the temporal shifts in gene expression networks that occur over the course of pre/postnatal neurodevelopment and provide insight into the biological processes that become inactivated in adulthood. While embryonic genes that become inactivated in late postnatal stages of development may be considered prime candidates as promoters of axon regeneration, mechanisms by which the capacity to extend neurons in the embryo ceases in the adult CNS remain elusive. Recent findings interrogating open chromatin regions at embryonic (E14) and postnatal (P2) stages in retinal progenitors revealed that the transcription factor Lhx2 is necessary for developmental transitions of open chromatin states across the genome[70]. Knockout assays showed that Lhx2 regulates both local and global chromatin accessibility for motifs of pioneer transcription factors, suggesting that developmentally vital transcription factors, such as Lhx2, promote developmental transitions via epigenomic interactions. These findings highlight the molecular transitions that occur across neurodevelopment and implicate the epigenome (e.g., DNA methylation) as a potential regulator of these temporal shifts.

**Putative role of DNA methylation in Na$^+$/K$^+$-ATPase regulation for RGC regeneration.** The Na$^+$/K$^+$-ATPase pump functions as a transmembrane enzyme that allows for the exchange of sodium and potassium ions across the lipid bilayer[97], and its relationship with axon regeneration has been previously noted, primarily in invertebrate species[61,98–100]. Expression of Na$^+$/K$^+$-ATPase subunits is highest during blastocyst and embryonic development[101], and is persistently expressed through early brain development[102]. Expression of these subunits declines with age in adult mammalian retinal tissue[103].

Conversely, in amphibians and fish that are known to retain the ability to regenerate injured central axons, Na$^+$/K$^+$-ATPase expression is maintained into adulthood[61,98–100]. For example, protein levels of the α1 ATPase subunit is significantly upregulated during retinal regeneration in adult newts[98]. Following optic nerve transection in goldfish, mRNA of the α3 ATPase subunit is highly expressed in the ganglion cell layer of regenerating retinas 2 days after axotomy, peaking at 5-10 days, and returning to baseline levels by day 45[100]. Furthermore, the β3 subunit (homolog to mammalian β2 and β3), exhibits increased mRNA expression in RGCs of adult zebrafish following axotomy-induced retinal regeneration[99], in line with our findings of increased β2 in IR RGCs. Moreover, the regenerative regions of the *Xenopus* CNS show coordinated Na$^+$/K$^+$-ATPase subunit transcription and DNA methylation abundance[64,104]. These include hypomethylation and increased expression of *atp1a1*, mirroring our findings of decreased *Atp1a1* methylation and increased expression in regenerated rat RGCs. Finally, in a rat model of peripheral nerve injury, a tissue that also displays robust regenerative capacity, the Na$^+$/K$^+$-ATPase subunits show unique protein expression profiles, including reduction of α1 and α2 and transient increase of α3 and β3, both of which diminish to baseline levels during regeneration[61]. These data indicate that the Na$^+$/K$^+$-ATPase is active in neural tissues with regenerative properties.

In this study, we show that several subunits of Na$^+$/K$^+$-ATPase (i.e., *Atp1a1, Atp1a2, Atp1a4, Atp1b1, Atp1b2, Atp1b3,* and *Atp1b4*) are differentially methylated in regenerating axons from adult mammalian RGCs, and at least one of these represents reactivation of embryonic growth signals (i.e., *Atp1b2*). In addition, these DNA methylation alterations correlate with Na$^+$/K$^+$-ATPase subunit transcript levels, suggesting modulations in DNA methylation levels may be driving the transcription of the Na$^+$/K$^+$-ATPase in the mammalian CNS to facilitate regeneration of RGCs. This hypothesis was validated when chemical inhibition of the Na$^+$/K$^+$-ATPase pump activity and heterozygous deletion of Na$^+$/K$^+$-ATPase in vivo (i.e., in the *Atp1a1*$^{+/-}$ and *Atp1a2*$^{+/-}$ mouse models) both significantly reduced the ability of adult RGCs to regenerate axons into peripheral nerve grafts. Furthermore, investigation of DNA sequences immediately flanking the differentially methylated sites annotated to Na$^+$/K$^+$-ATPase subunits revealed that 12 of the 14 flanks contained putative binding motifs for transcription factors that are significantly enriched among the DMR sequences. These findings further suggest that DNA methylation plays a functional role in mature mammalian RGCs after injury, likely influencing the binding of transcription factors and altering the expression of the Na$^+$/K$^+$-ATPases. While the mechanism by which the Na$^+$/K$^+$-ATPases govern mammalian CNS regeneration is still largely unexplored, the role of Na$^+$ and K$^+$ gradients in providing the neuronal action potentials needed for cell proliferation, differentiation, and migration, as well as axon guidance and outgrowth, and other regenerative activities, is well established[105].

Using a novel method to select and purify regenerated from non-regenerated neurons, the data presented here show large-scale genome-wide changes in the molecular makeup of regenerated axons, and highlight a critical role for the Na$^+$/K$^+$-ATPase in mammalian CNS axon regeneration through DNA methylation.

## Methods

**Animals.** All animal housing and surgical procedures were conducted following the Guide for the Care and Use of Laboratory Animals and under conditions approved by The Research Animal Resources and Care Committee of the University of Wisconsin (M005286). Adult male Sprague-Dawley rats were obtained from

Harlan Laboratories, Madison, WI, USA. Rats were housed in three per cage under a standard 12 hours light/12 hours dark cycle, with food and water *ad libitum*. Heterozygous α-1 and α-2 mice were obtained from Jerry Lingrel, PhD (University of Cincinnati).

### Rat experiments

#### Tissue collection

Embryos: On days 16–18 of pregnancy, dams (rats) were anesthetized with Ketamine (40–80 mg/kg) and Xylazine (5–10 mg/kg) before euthanasia. All embryos were removed and the embryonic retinas harvested, RGCs dissociated, and the DNA extracted for whole-genome bisulfite sequencing.

Adult animals: Six-week-old male rats that served as control animals in a different study (not reported here), and which had been treated with daily intraperitoneal injections of double deionized water (DDI), were anesthetized with Ketamine (40–80 mg/kg) and Xylazine (5–10 mg/kg) intraperitoneally. An extra 10–20 mg/kg of Ketamine was given if needed, and surgery was performed as previously described[52]. The skin was opened over the midline of the skull and the left orbit was approached laterally. The left optic nerve was sectioned within 2 millimeters of the globe and injury to the retinal artery was avoided. A 2-centimeter sciatic nerve segment was harvested from the left hind limb and grafted at the optic stump using 10-0 nylon sutures (Ethicon, San Angelo, TX). The end of the graft was allowed to lie freely in the subcutaneous layer over the skull and secured with a 4.0 black braided silk suture (Ethicon), after which the incision was closed. Eight weeks later, the animal was sedated, the wound reopened, and the free end of the sciatic graft ~2 centimeters from the globe was identified, cut sharply, and a gelfoam soaked in 5 μl of 50 mg/ml of Oregon Green (cat#D7172, Invitrogen) was placed at the cut site. Forty-eight hours later, under sedation, both the injured and uninjured (contralateral) retinas were harvested, the RGCs dissociated, and the DNA extracted for whole-genome bisulfite sequencing.

### Retina dissociation

For downstream experiments and analyses, three biological replicates were used for each treatment group. Four retinas were pooled for each biological replicate. The eye globe was collected in cold HBSS media (without $Ca^{2+}$ and $Mg^{2+}$; cat#14170-112 Invitrogen) and the retina was dissected from embryos and adult animals. Following dissection, the media containing each retina was transferred to a 15 ml conical tube containing 3.5 ml of enzyme solution (BSA 7.5% (cat#A8412, Sigma), cysteine, 0.42 mg/ml (cat#C2529), NaOH (1.2 mM, cat#S2770, Sigma and papain (4 mg/ml; cat#LS03119, Worthington Biochemical Corporation). An additional 250ul of papain was added to the 15 ml conical tube and the solution was incubated at 37 °C for 37 minutes with gentle rocking. The supernatant was then aspirated without disturbing the retina, and the retina incubated again with 3.5 ml of enzyme and 250 μl of papain. Following incubation and aspiration, 5 ml of cold HBSS was added to the undisturbed retina, and the tube was inverted to mix, and the supernatant was aspirated. Then, 5 ml of 0.1 M PBS (DEPC) with 1% glucose + DNse was added to the 15 ml conical tube containing the retina, and the tube was inverted and incubated upright for 7 minutes at room temperature. Following this incubation, 2 ml of the supernatant was aspirated and the remaining supernatant was gently mixed with 5 ml serological pipette 15 times followed by gentle mixed 50 times with 1 ml serological pipette. The mixed solution was transferred to a 50 ml tube through a 40 μm Nylon cell strainer (cat#410-001-OEM, Foxx) and then to a 5 ml polystyrene tube. The cells were then washed with ice-cold FACS buffer (0.1 M PBS (DEPC), PBS + 2%FBS at a concentration of 1 ml $1 \times 10^6$ cells/ml), centrifuged at $500 \times g$ for 5 minutes, then the supernatant was aspirated and the cells were re-suspended in 500 μl of FACS buffer and transferred into a flow tube with 1ul of neuronal Thy-1 antibody (cat#554898, BD Parmingen) conjugated to R-Phycoerythrin (Thy-1 PE), and incubated at 4 °C for 1 hour. The cells were then washed in ice-cold FACS buffer, centrifuged at $500 \times g$ for 7 minutes, and the supernatant was aspirated and the cells were re-suspended in PBS (DEPC). One milliliter of the solution was processed by FACS.

### Fluorescence-activated cell sorting

Cells were initially gated based on the relative size and granularity using forward scatter and side-scatter to separate cells from debris, then single cells were gated on forward scatter area versus forward scatter height, followed by a second singlet gate using side-scatter area versus side-scatter height. Live single cells were then gated as DAPI negative. Finally, cells were gated into groups based on Thy-1 PE expression and Oregon Green fluorescence. Specifically, RGCs that regenerated axons (i.e., their axons reached the end of the sciatic nerve graft and were backfilled with Oregon Green) were Thy-1 positive/Oregon Green positive. Conversely, RGCs that did not regenerate axons after optic nerve transection, or those obtained from the embryonic or adult uninjured retinas, were Thy-1 PE-positive/Oregon Green negative.

### FACS-sorted genomic DNA methylation

Live sorted cells were collected in 200 μl of DNA/RNA shield (cat#R1100, Zymo Research), and genomic DNA was extracted using the Quick DNA Miniprep Plus Kit (Cat#D4068, Zymo Research) following the manufacturer's protocol. Quantity and quality of extracted genomic DNA were determined by Aligent 2100 DNA Bioanalyzer (Biotech Center, UW-Madison). Fifty nanograms of genomic DNA was sodium bisulfite-treated to

convert unmethylated cytosines to uracils, and sequencing libraries were prepared using the Pico Methyl-Seq Library Prep Kit (cat# D5455, Zymo Research). The converted libraries were purified and prepared for whole-genome sequencing at the Roy J. Carver Biotechnology Center (Urbana, IL), generating an average of 791 million 150 bp reads per sample ($N = 3$ biological replicates; 4 retinas per replicate).

### Whole-genome bisulfite sequencing data analysis

The quality of raw FASTQ files was assessed using FastQC. Quality trimming of reads was performed using trim_galore[106] using the following parameters: 4 bp were trimmed from the 5' end of each read, minimum length of reads was capped at 50 basepairs, a quality score of 30 employed, and Ns were trimmed from each read. Following trimming, reads were aligned to the UCSC *rn6* rat genome using Bismark[107], coupled with bowtie2[108], using the following parameters: reads were aligned in non-directional mode, the minimum score was set to L,0,-0.4, and unmapped reads were extracted. Unmapped reads were trimmed again using trim_galore; however, 80 basepairs from the 3' end of the unmapped reads were removed. Trimmed unmapped reads were realigned to the genome using similar parameters as above. Following alignment, all reads were concatenated to one file and deduplicated using Bismark. Methylation extraction and reporting of the methylation level from a single CpGs was performed using Bismark.

### Identification of differentially methylated regions and loci

R package *DSS*[109] was used to identify differentially methylated regions (DMRs) and loci (DMLs). CpGs were removed if their coverage was <10×, leaving ~13 million CpGs tested for each sample. Differentially methylated loci (DMLs) were identified by estimating dispersion and smoothing methylation levels over a 500 basepair window using *DSS*. DMLs were called for the comparisons of injured/regenerated (IR) *vs.* injured/non-regenerated (INR), IR *vs.* uninjured adult (UA), and INR *vs.* UA using the following parameters: FDR < 0.01, differential methylation >20%. Considering the extensive molecular differences between embryonic and adult cells, more stringent statistical cutoffs were applied to identify DMLs for the comparisons of IR *vs.* uninjured embryonic (UE), UE vs. UA, and INR *vs.* UE using the following parameters: FDR < 0.01, differential methylation >50%. For the three comparisons of IR *vs.* INR, IR vs. UA, and INR *vs.* UA, the following parameters were used for DMR calling: a minimum of 10% difference in methylation between experimental groups, a *P*-value threshold of 1e-4 was used to determine DMLs, a minimum of three DMLs were located within the region, and 50% of CpGs in the region were significantly differentially methylated. Considering the extensive molecular differences between embryonic and adult cells, more stringent statistical cutoffs were also applied to identify DMRs for the comparisons IR v UE, UE v UA, and INR v UE using the following parameters: a minimum of 20% difference in methylation between experimental groups, a *P*-value threshold of 1e-4 was used to determine DMLs, a minimum of five DMLs were located within the region, and 99% of CpGs in the region were significantly differentially methylated. DMRs and DMLs from all six two-way comparisons of the four experimental groups were identified using these parameters. DMRs/DMLs were annotated to genes and genomic features using R package ChIPseeker[110], using 10,000 basepairs and 3,000 basepairs upstream and downstream, respectively, of the transcription start site as the promoter region. R packages *ChIPseeker* (v1.32.0), *GenomicRanges* (v1.48.0), *org.Rn.eg.db*(v3.15.0), and *TxDb.Rnorvegicus.UCSC.rn6.refGene* (v3.4.6) were used for annotation. Gene ontologies were performed using R package *clusterProfiler*[111]. The Hypergeometric Optimization of Motif EnRichment (HOMER) toolkit was used to identify enriched transcription factor motifs using sequences from the *rn6* rat genome as background set, using a *q* value <0.01 to identify significant transcription factors[112]. Notably, for motif analysis of DMLs associated with $Na^+/K^+$-ATPase subunits ($N = 24$), genomic sequences immediately flanking the DML (+/−125bp) were extracted from the rat genome. As several of these extended regions overlapped, *GenomicRanges*[113] was used to combine and reduce these redundant coordinates/regions, leaving $N = 14$ regions for further analysis.

### Reverse transcription quantitative PCR

Reverse transcription quantitative (RT)-qPCR was performed with 1 μg of total RNA from each experimental group (INR and IR), which was reverse transcribed with oligo(dt)$_{15}$ and random hexamer primers using M-MuLV reverse transcriptase (Life Technologies, Rockville, MD). Ten nanograms of cDNA and gene-specific primers were added to SYBR Green PCR Master Mix (SYBR Green I Dye, Ampli Taq® DNA polymerase, dNTPs, with dUTP and optimal buffer components; Applied Biosystems, Foster City, CA) and subjected to PCR amplification (1 cycle at 50 °C for 2 min, 1 cycle at 95 °C for 10 min, and 40 cycles at 95 °C for 15 sec and 60 °C for 1 min) in a TaqMan 5700 Sequence Detection System (Applied Biosystems). For each transcript, RT-PCR was conducted in triplicate using RNA from RGC samples from different sets of animals. The amplified transcript was qualified with the comparative $C_T$ method using 18 s rRNA as the internal control. The following primer sequences (designed using the Primer Express software; Applied Biosystems) were used.

α1: AGCACATGATGCCTCCAGAGA (forward); ATCTGGATCTGCCCGTC ACT (reverse); β1: AAGAGCACTGTACTCCTGGCAATC (forward); GGTGTTAAATGTTGTCTACAGTGCAA (reverse);

*Negr1*: GGTGACAAGTGGTCAGTGGA (forward); CATCGTCCGTGGTGTG TGTT (reverse);

*Cd38*: CCAGATCGGTCTTGGAGTGG (forward); CGTGGTAGGCTTCCCTT TCC (reverse);

*Nav3*: GTTCACGCTGCTCTTCCGAT (forward); CGCTCCGGTGTAGCAA ATTC (reverse);

*Il4r*: TGCACCAAGTTCCTGTCCTC (forward); ACTCACACGTAGAAGTG CGG (reverse);

*Rassf4*: AAGGACTCCTCAACATCGCC (forward); GTGCACTCGTTCCCGA TCAT (reverse);

*Nefl*: CGCAAGAAGATGGATGAGCC (forward); GGGTGGAGGTAACCGA TAGC (reverse);

*Arhgap20*: AGGCATGTCTCGCATCGC (forward); TAAGGCTTTGTCCAGG ATGAGG (reverse).

**Immunoblot analyses**. Immunoblot analyses were performed as previously described[114]. Briefly, the cells were lysed in lysis buffer (Sigma) and the protein content was determined by Lowry's method[115]. Protein samples (30 μg/lane) and pre-stained molecular mass markers (Bio-Rad, Hercules, CA) were denatured in SDS reducing buffer (1:2 by volume, Bio-Rad). The samples were then electrophoretically separated on 12.5% Tris-HCl gels (Biorad) and the resolved proteins were electrophoretically transferred to a polyvinylidene difluoride membrane (PVDF, 0.2 μm; Bio-Rad), which was then incubated in 5% non-fat dry milk in Tris-buffered saline (TBS) for 30 minutes. The membrane was then incubated overnight at 4 °C with a mouse monoclonal antibody raised against the Na$^+$/K$^+$-ATPase α-subunit (1:200; Santa Cruz Biotechnology, Inc., Santa Cruz, CA) or with a mouse monoclonal antibody raised against the Na$^+$/K$^+$-ATPase β-subunit (1:500; Santa Cruz). The membrane was rinsed with TBST and incubated with corresponding horseradish peroxidase-conjugated secondary IgG (1:2000; Santa Cruz) for 1.5 hour at room temperature. Bound antibody was visualized using the enhanced chemiluminescent solution (Pierce, Thermo Fisher Scientific, Rockford, IL) as per the manufacturer's instructions. The chemiluminescent signal was captured on autoradiographs (Eastman Kodak, Rochester, NY), which were scanned and the intensity of the autoradiograph signals (including a blank region) was determined using the NIH ImageJ Software. Each immunoblot was then stripped using stripping buffer (Pierce) for 30 minutes and re-probed with a mouse monoclonal antibody against β-actin (1:4000; Sigma) and the expression of β-actin was quantitated as described above. Control and treatment values were corrected for blank values, normalized to their respective β-actin band intensity, and the results were then expressed as mean ± standard error of the mean (SEM).

**Digoxin-mediated inhibition of the Na$^+$/K$^+$-ATPase activity**. 15 adult male Sprague-Dawley rats were used for Na$^+$/K$^+$-ATPase activity experiments, seven were used as control, and eight were treated with digoxin (Sigma-Aldrich Corp., St. Louis, MO, USA). Digoxin was dissolved in DMSO and diluted in PBS to 1% DMSO for injection. Rats were treated with digoxin daily (1 mg/kg), administered intraperitoneally. Control rats were similarly treated with 1% DMSO. Injections were started three days preoperatively and continued for two weeks post-operatively. For the optic nerve regeneration model, the optic nerve was exposed through a lateral orbital approach and cut within 2 mm of the globe, and connected to a sciatic nerve. After 2 months, Fluorogold tracer was used to label regenerated neurons following backfilling of the nerve. After 48 hours, the retina was harvested and postfixed, and the flat-mount was analyzed for fluorescence under a microscope. An observer blinded to the treatment conditions recorded the number of RGCs labeled with Fluorogold.

**Na$^+$/K$^+$-ATPase knockout mice: genotyping and surgery**. Heterozygous α-1 females were bred with heterozygous α1 males, and the heterozygous α2 females were bred with heterozygous α2 males. The resulting offspring were genotyped by Southern blot, as previously described[116]. For analysis of RGC regeneration in vivo, adult (>4 weeks of age) male mice with the following genotype: α1$^{+/-}$ ($n = 7$), α1$^{+/+}$ ($n = 6$), α2$^{+/-}$ ($n = 10$), and α2$^{+/+}$ ($N = 10$) were anesthetized with Keta-mine (100 mg/kg) and Xylazine (10 mg/kg). Only heterozygous and wild-type mice were used in our experiments as the homozygous knockout mice of either isoform were embryonic lethal[117]. Surgical procedures were similar to those described in rats above, as previously published[52,118]. Briefly, the right optic nerve was sectioned within one-millimeter of the globe and a one-centimeter sciatic nerve segment was harvested from the left hind limb and grafted at the optic stump using 11-0 nylon sutures. After two months, 5 μl of the retrograde fluorescent tracer Fluorogold 5% (Fluorochrome, LLC, Denver, Colorado) was placed at the free cut end of the graft. After 48 h, the animals were sedated, the retina was removed and post-fixed, and the flat mount was analyzed for fluorescence (regenerated cells) under a microscope.

**Statistical analysis of immunoblots and RT-PCR data**. Statistical analyses between control and treatment groups were performed using the Student's *t* test or analysis of variance, as appropriate. A *P* value <0.05 was considered statistically significant. For each of the flow cytometry experiments, three biological replicates were used, each comprising four dissociated retinas that were pooled together (total $N = 12$ retinas).

**Reporting summary**. Further information on research design is available in the Nature Portfolio Reporting Summary linked to this article.

## Data availability

Sequence data have been submitted to Gene Expression Omnibus, with the accession number TBD.

## Code availability

Scripts and code used for the generation of data, results, and figures used in this manuscript have been deposited at https://github.com/andymadrid/RGC_DNAm.git. Video files can be viewed at https://uwmadison.box.com/s/mli3znh1av9coj67tw59si10se6v6ynr.

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

## Acknowledgements

We acknowledge the UWCCC Flow Cytometry Laboratory for performing the cell sorting and providing experimental consultation. We would like to thank Dr. Carolina D. Alberca and Ms. Isabelle Renteria for their assistance in preparing manuscript figures. This work was supported by funds from the National Institute of Child Health and Human Development (1R01HD047516) (B.J.I.), The University of Wisconsin Medical School Research Committee (B.J.I.), The American College of Surgeons Faculty Research Fellowship (B.J.I.), the University of Wisconsin Department of Neurological Surgery R&D (R.A. & B.J.I.), and the University of Wisconsin Carbone Cancer Center Support Grant (P30 CA014520).

## Author contributions

B.J.I. and R.S.A. conceived the experiments. E.R., J.K., D.S., K.S., D.C., S.L., F.H., L.A.P., and N.H. performed animal and molecular experiments. A.M. performed statistical analyses of data and generated figures. E.R., A.M., R.S.A., and B.J.I. wrote the manuscript. E.R., A.M., L.A.P., N.H., R.S.A., and B.J.I. edited the manuscript.

## Competing interests

The authors declare no competing interests.
