## [Peer Review File · Communications Biology]

Reviewers' comments:

Reviewer #1 (Remarks to the Author):

The manuscript titled "Purified neuronal populations of regenerating retinal ganglion cells reveal DNA methylation-mediated role of Na⁺/K⁺-ATPase in axon regeneration" described the intrinsic properties of regenerating RGC vs non regenerating RGC. In this article author develop a new method to identify regenerative RGC and non-regenerative RGC using sciatic nerve graft. Author compares DNA methylated region of regenerative RGC to non-regenerative RGC and they identified important genes methylated during the regeneration condition. Manuscript was professionally written and well explained with few exceptions that will be improve after minor revisions

Recommendation: publish after minor revision

Comments

1. In introduction section line 61-62 authors discussed the role of Schwann cells and oligodendrocytes but not satellite glial cells while growing evidence suggests that Satellite Glial cell also play crucial role in axonal growth so, please add some more relevant references related to satellite glial cell such as Avraham et al. Nature Com. 2020
2. In method section, if author explain RGC tracing method in a one place, it would be good for reader.
3. Author claiming, in line 296-297, "We developed a Fluorescence-Activated Cell Sorting (FACS) following optic nerve transection in rats" in my understanding FACS sorting is normal they only develop protocol for tracing regenerative RGC using sciatic nerve graft. It would be good if author give more attention to explain the novelty of protocol.
4. I am curious about the tracer, why author only use Oregon green? they tried other tracer such as cholera toxin B and dextran.
5. In result section line 424-432, author written "Inhibition of Na⁺/K⁺-ATPase activity axonal regeneration" also in text author talking about axonal regeneration while in figure 6C only present RGC number (RGC survival) not axonal regeneration, kindly use appropriate terminology (RGC survival) according to data and method.
6. In Result section line 434-444 again author use axon regeneration in text while in data fig 6D &E author have only presented RGC number, kindly use RGC survival in text.
7. In figure 6 C D E only has RGC quantification data, author need to incorporate Fluoro-Gold staining image in main figure
8. In figure legend 6C author written- "The number of regenerated axonsare depicted" while in data they shown RGC count not axon please correct it.
9. Do the author check the axonal regeneration status in optic nerve with digoxin-treated or in heterozygous knockout $\alpha 1$ & $\alpha 2$ mice, if yes please provide the axonal regeneration data.
10. Discussion was well written except the section- Role of embryonic molecular signalsadult RGCs, is poorly discussed without any relevant reference. If author explain more with good example such as KP Gill et al. Sci Report 2016, W Pita-Thomas et al. Sci Report· 2021 and NM Tran et al. Cell 2019 it would be good for audience

Reviewer #2 (Remarks to the Author):

The authors in this study have developed a cell sorting method to segregate retinal ganglion cells (RGCs) and carried out WGBS profiling of these RGCs to identify genes and pathways linked to mammalian RGC regeneration. They have also reported Na⁺/K⁺-ATPase subunits play a role in embryonic and mature RGC axon growth and regeneration.

However, the manuscript has multiple issues in its current form and there are some points that need to be considered to improve this work.

More specific comments are given below.

- The experimental design is not clear for the entire study. The authors should include a workflow as figure 1 or in supplementary figure to explain the entire study design.
- The authors need to describe every method used in the study in detail
- The authors need to mention how many male and female rat retinas were used for whole genome bisulfite sequencing for adult and embryonic stage

- What was the read length for the WGBS data and please mention it in the methods
- Which version of the rat genome was used for alignment for WGBS data
- DMLs were called for the comparisons IR, INR, UA, and UE groups. What is IR, INR, UA, UE the authors need to define these groups in methods.
- In Figure 2D the authors have explained the annotation of DMRs in the genome and majority of them are in intergenic region, but they have not mentioned in the methods how was the annotation done and what was the source of the annotation.
- In the figures the authors should add figure number on the right-side top region as it is difficult to track the figures
- From the 48 genes shared between adult and embryonic axon that showed growth potentials, how did the author choose Na⁺/K⁺-ATPase subunit gene Atp1b2 gene and what about the other 47 genes. The authors need to describe in detail in results how and why were Na⁺/K⁺-ATPases chosen for validation and were there any other candidate's genes in those 47 genes.

Minor comments

In Figure 2D the label on x-axis and y-axis are not clearly readable
The fonts in all the figures and supplementary figures are very small

Response to Reviewers

Article: Purified neuronal populations of regenerating retinal ganglion cells reveal DNA methylation-mediated role of Na⁺/K⁺-ATPase in axon regeneration

Corresponding author: Bermans J. Iskandar (iskandar@neurosurgery.wisc.edu)

We appreciate the positive review to our manuscript entitled: *Purified neuronal populations of regenerating retinal ganglion cells reveal DNA methylation-mediated role of Na⁺/K⁺-ATPase in axon regeneration*. The following are our point-by-point responses to the reviewers' comments:

Editor's comments:

1. (From Reviewer 2): *In particular, we will need you to include a figure describing the experimental workflow and a clear explanation of the whole experimental set up.*

Authors' Response: Thank you for the suggestion. In response, we now provide an illustrated flowchart (Fig. 1) that describes the overall methodology to generate IR, INR, UA, and UE cells. The flowchart is accompanied by 3 videos, one for each method.

Change to Text: See Videos 1-3 and Figure 1

2. (From Reviewer 1): *Additionally, you would need to provide a more detailed description of the RGC tracing method and highlight the novelty of the method proposed in the manuscript.*

Authors' Response: Done.

Change to Text: Refer to answer above with regard to the RGC tracing method.

Discussion: Until now, studies profiling regeneration signals in the optic nerve had relied on a comparison between injured (usually optic nerve crush or transection) and uninjured RGCs, without specifically segregating injured neurons that regenerated from those that have not. A similar FACS approach was used to separate retinal cells based on cell type (e.g., RGC vs. non-RGC) and injury state (Fischer, Petkova et al. 2004, Hartl, Krebs et al. 2017), while others separated the neurons based on their stage of development (Zibetti, Liu et al. 2019). Molecular signatures of CNS tissues that are known to regenerate after injury (frog eye and tadpole hindbrain) were compared to others that are incapable of such regeneration (e.g., frog hindbrain) (Reverdatto, Prasad et al. 2022). And more recently, RGC sorting was done by profiling cells based on genome-wide loss-of-function screen for factors limiting axonal regeneration after optic nerve crush (Lindborg, Tran et al. 2021). However, this complex method identified genes that inhibit, but not those that enhance, axon regeneration. This and other single-cell RNA sequencing technologies and transcription factor screens have yet to provide researchers with the ability to profile cells based on their proven ability to regenerate axons in vivo, underscoring both the novelty and relative simplicity of our method (Norsworthy, Bei et al. 2017, Lindborg, Tran et al. 2021, Yang, Jian et al. 2021).

Reviewer 1's comments:

1. In introduction section line 61-62 authors discussed the role of Schwann cells and oligodendrocytes but not satellite glial cells while growing evidence suggests that Satellite Glial cell also play crucial role in axonal growth so, please add some more relevant references related to satellite glial cell such as Avraham et al. Nature Com. 2020

Authors' Response: We thank the reviewer for the insight and have provided to the following addition into the Introduction section of our manuscript to highlight the growing evidence of satellite glial cells in regenerative growth.

Change to Text: “Moreover, a growing body of research has found that satellite glial cells, a subpopulation of glial cells found in the peripheral nervous system, promote regenerative growth in peripheral neurons following injury (Avraham, Deng et al. 2020, Jager, Pallesen et al. 2020, Avraham, Feng et al. 2021), further underscoring the importance of glial cell recruitment and function in recovery in the mammalian nervous system.”

2. In method section, if author explain RGC tracing method in a one place, it would be good for reader.

Authors' Response: See response above.

Change to Text: Flowchart (Fig. 1) and Videos 1-3, as above.

3. Author claiming, in line 296-297, “We developed a Fluorescence-Activated Cell Sorting (FACS) following optic nerve transection in rats” in my understanding FACS sorting is normal they only develop protocol for tracing regenerative RGC using sciatic nerve graft. It would be good if author give more attention to explain the novelty of protocol.

Authors' Response: Details and comment on novelty were added to the text.

Change to Text: See Results (Paragraph 1) and Conclusions (Paragraph 1): Until now, studies profiling regeneration signals in the optic nerve had relied on a comparison between injured (usually optic nerve crush or transection) and uninjured RGCs, without specifically segregating injured neurons that regenerated from those that have not. A similar FACS approach was used to separate retinal cells based on cell type (e.g., RGC vs. non-RGC) and injury state (Fischer, Petkova et al. 2004, Hartl, Krebs et al. 2017), while others separated the neurons based on their stage of development (Zibetti, Liu et al. 2019). Molecular signatures of CNS tissues that are known to regenerate after injury (frog eye and tadpole hindbrain) were compared to others that are incapable of such regeneration (e.g., frog hindbrain) (Reverdatto, Prasad et al. 2022). And more recently, RGC sorting was done by profiling cells based on genome-wide loss-of-function screen for factors limiting axonal regeneration after optic nerve crush (Lindborg, Tran et al. 2021). However, this complex method identified genes that inhibit, but not those that enhance, axon regeneration. This and other single-cell RNA sequencing technologies and transcription factor screens have yet to provide researchers with the ability to profile cells based on their proven ability to regenerate axons in vivo, underscoring both the novelty and relative simplicity of our method (Norsworthy, Bei et al. 2017, Lindborg, Tran et al. 2021, Yang, Jian et al. 2021).

4. *I am curious about the tracer, why author only use Oregon green? they tried other tracer such as cholera toxin B and dextran.*

Authors' Response: It is true that nowadays many tracers are available. Oregon green has been successfully used for a number of years in our laboratory, has consistently shown very stable signal generation in flow cytometry (as used in this manuscript), and has an excitation range that does not overlap with R-Phycoerythrin (red), thus allowing clean separation on flow cytometry.

Change to Text: None

5. *In result section line 424-432, author written "Inhibition of Na⁺/K⁺-ATPase activity axonal regeneration" also in text author talking about axonal regeneration while in figure 6C only present RGC number (RGC survival) not axonal regeneration, kindly use appropriate terminology (RGC survival) according to data and method.*

Authors' Response: This does represent the number of RGCs that regenerated axons, not the ones that survived. Specifically, these are the injured neurons whose axons grew to the end of the graft, allowing them to take up the fluorescent dye and transport it to the cell body.

Change to Text: The flowchart (Fig. 1) and Videos 1-3 clarify the method used.

6. *In Result section line 434-444 again author use axon regeneration in text while in data fig 6D &E author have only presented RGC number, kindly use RGC survival in text.*

Authors' Response: As above, these represent the number of RGCs that regenerated axons, not those that survived.

Change to Text: The flowchart (Fig. 1) and Videos 1-3 clarify the method used.

7. *In figure 6 C D E only has RGC quantification data, author need to incorporate Fluoro-Gold staining image in main figure*

Authors' Response: Since a Fluorogold-stained retina flat mount is shown in Fig. 1C, we added a sentence at the end of the Fig. 6 legend that refers to the photomicrograph, which is also shown in Video 1.

Change to Text: Fig. 6 (end): Fig. 1C shows a representative image (20x) of retina flat mount with the fluorescent (Fluorogold-positive) RGCs that regenerated axons to the end of the graft. The fluorescent cells are counted under fluorescent microscopy to generate the data in Figs. 6C-E (Video 1).

8. *In figure legend 6C author written- "The number of regenerated axonsare depicted" while in data they shown RGC count not axon please correct it.*

Authors' Response: Done

Change to Text: The new sentence states: The number of fluorescent RGC neurons (i.e., those with regenerated axons).

9. Do the author check the axonal regeneration status in optic nerve with digoxin-treated or in heterozygous knockout $\alpha 1$ & $\alpha 2$ mice, if yes please provide the axonal regeneration data.

Authors' Response: We do not look for regeneration in the optic nerve, only in the retina via the retrograde tracers that is instilled in the regenerated axons.

Change to Text: None

10. Discussion was well written except the section- Role of embryonic molecular signalsadult RGCs, is poorly discussed without any relevant reference. If author explain more with good example such as KP Gill et al. Sci Report 2016, W Pita-Thomas et al. Sci Report- 2021 and NM Tran et al. Cell 2019 it would be good for

Authors' Response: We thank the reviewer for their suggestion. To expand and make our Discussion section more complete, the subtopic noted by the reviewer has been amended to include more discussion points from previous research articles.

Change to Text: Discussion: Role of Embryonic ...: Previous research discerned extensive differences between the transcriptomes of human embryonic stem cells (hESCs) and those of hESC-derived RGCs (Gill, Hung et al. 2016), with strong similarities between hESC-derived RGCs and adult human RGCs. Moreover, open chromatin regions enriched in embryonic cells were found to harbor binding motifs for transcription factors with potential roles in axon growth (e.g., CREB) of postnatal RGCs. In turn, overexpression of CREB fused to the VP64 transactivation domain in RGCs was found to promote axonal regeneration following optic nerve injury (Pita-Thomas, Gonçalves et al. 2021), confirming that reactivation of embryonic signals may enhance axonal regeneration in adult neurons. Recent research, using single-cell RNA-sequencing technologies, have investigated the temporal changes in gene expression across embryonic and postnatal development (Whitney, Butrus et al. 2022). Using 3 embryonic (E13, 14, and 16) and 3 postnatal (P0, 5, and 56) retinas, researchers identified 6 sets of genes with altered expression across embryonic and postnatal development, shifting from gene network programs primarily associated with axon guidance (early embryonic module) to synaptic refinement processes (late postnatal module). These results highlight the temporal shifts in gene expression networks that occur over the course of pre/postnatal neurodevelopment and provide insight into the biological processes that become inactivated in adulthood. While embryonic genes that become inactivated in late postnatal stages of development may be considered prime candidates as promoters of axonal regeneration, mechanisms by which the capacity to extend neurons in the embryo ceases in the adult CNS remain elusive. Recent findings interrogating open chromatin regions at embryonic (E14) and postnatal (P2) stages in retinal progenitors revealed that the transcription factor Lhx2 is necessary for developmental transitions of open chromatin states across the genome (Zibetti, Liu et al. 2019). Knockout assays for Lhx2 showed that it regulates both local and global chromatin accessibility for motifs of pioneer transcription factors, suggesting that developmentally vital transcription factors, such as Lhx2, promote developmental transitions via epigenomic interactions. These findings highlight the molecular transitions that occur across neurodevelopment and implicate the epigenome (e.g., DNA methylation) as a potential regulator of these temporal shifts.

Reviewer 2's comments:

1. *The experimental design is not clear for the entire study. The authors should include a workflow as figure 1 or in supplementary figure to explain the entire study design.*

Authors' Response: Done.

Change to Text: See flowchart (Fig. 1) and Videos 1-3, as above.

2. *The authors need to describe every method used in the study in detail*

Authors' Response: Done

Change to Text: See flowchart (Fig. 1) and Videos 1-3, as above.

3. *The authors need to mention how many male and female rat retinas were used for whole genome bisulfite sequencing for adult and embryonic stage*

Authors' Response: All animals were male. There are 3 groups used as biological replicates. Each group consists of 4 retinas that are pooled together.

Change to Text: $N = 12$ retinas (clarified in methods and legends)

4. *What was the read length for the WGBS data and please mention it in the methods*

Authors' Response: The read length of the raw sequenced data was 150bp, prior to trimming. We have added this information to the Methods.

Change to Text: The converted libraries were purified and prepared for whole-genome sequencing at the Roy J. Carver Biotechnology Center (Urbana, IL), generating an average of 791 million 150bp reads per sample.

5. *Which version of the rat genome was used for alignment for WGBS data*

Authors' Response: The UCSC *rn6* version of the rat genome was used for alignment of the bisulfite-treated reads.

Change to Text: Reported in methods.

6. *DMLs were called for the comparisons IR, INR, UA, and UE groups. What is IR, INR, UA, UE the authors need to define these groups in methods.*

Authors' Response: IR, INR, UA, and UE are abbreviations for injured regenerated, injured non-regenerated, uninjured adult, and uninjured embryonic neurons, respectively.

Change to Text: We added clarifications in the text as well as the new videos (Videos 1-3) and flowchart (Fig. 1). Additionally, we have added a section in which we list frequently used abbreviations:

ABBREVIATIONS

1. RGC (retinal ganglion cell)
2. IR (injured/regenerated)
3. INR (injured/non-regenerated)
4. UA (uninjured adult)

5. UE (uninjured embryonic)
6. DMR (differentially methylated region)
7. DML (differentially methylated loci)
8. CNS (central nervous system)

7. *In Figure 2D the authors have explained the annotation of DMRs in the genome and majority of them are in intergenic region, but they have not mentioned in the methods how was the annotation done and what was the source of the annotation.*

Authors' Response: For annotation of DMRs and DMLs to genomic structures and genes, we used a combination of R packages that are used for genomic region annotation. These included: *ChIPseeker* (v1.32.0), *GenomicRanges* (v1.48.0), *org.Rn.eg.db* (v3.15.0), and *TxDb.Rnorvegicus.UCSC.rn6.refGene* (v3.4.6). The Methods section has been amended to reflect this information.

Change to Text: Methods: Identification of differentially methylated regions and loci: R packages *ChIPseeker* (v1.32.0), *GenomicRanges* (v1.48.0), *org.Rn.eg.db*(v3.15.0), and *TxDb.Rnorvegicus.UCSC.rn6.refGene* (v3.4.6) were used for annotation.

8. *In the figures the authors should add figure number on the right-side top region as it is difficult to track the figures*

Authors' Response: Done
Change to Text: See figures

9. *From the 48 genes shared between adult and embryonic axon that showed growth potentials, how did the author choose Na⁺/K⁺-ATPase subunit gene *Atp1b2* gene and what about the other 47 genes. The authors need to describe in detail in results how and why were Na⁺/K⁺-ATPases chosen for validation and were there any other candidate's genes in those 47 genes.*

Authors' Response: Of all 48 genes that are shared between adult and embryonic tissues, the Na⁺/K⁺-ATPase family of genes is the one most associated with axon regeneration in the literature. This is discussed in detail in the Discussion section.

Change to Text: Results: In section entitled: DNA methylation levels in adult injured RGCs mirror those of uninjured embryonic RGCs: Of these, the Na⁺/K⁺-ATPase is the family of genes most linked to axon regeneration mechanisms in the literature (Arteaga, Gutiérrez et al. 2004, Bosse, Hasenpusch-Theil et al. 2006, Chen, Wang et al. 2011, Ellman, Isaksen et al. 2017, Tu, Katano et al. 2017, Lu, Shan et al. 2022, Reverdatto, Prasad et al. 2022). Specifically, the Na⁺/K⁺-ATPase subunit gene *Atp1b2* displayed hypermethylation in both growing UE RGCs and regenerating adult RGCs, implying that the Na⁺/K⁺-ATPase may reactivate its embryonic DNA methylation levels following injury to facilitate axon regeneration.

Discussion: See section entitled: **Putative role of DNA methylation in Na⁺/K⁺-ATPase regulation for RGC regeneration**

Minor comments

10. In Figure 2D the label on x-axis and y-axis are not clearly readable

Authors' Response: The font size used for the axis in the indicated figure have been enlarged.

Change to Text: See figure.

11. The fonts in all the figures and supplementary figures are very small

Authors' Response: The font size used for the axis in the indicated figure have been enlarged, where applicable.

Change to Text: See figure.

Sincerely,

Bermans J. Iskandar, MD and Andy Madrid, PhD
Department of Neurological Surgery
University of Wisconsin
Madison, WI

Arteaga, M. F., R. Gutiérrez, J. Avila, A. Mobasheri, L. Díaz-Flores and P. Martín-Vasallo (2004). "Regeneration influences expression of the Na⁺, K⁺-atpase subunit isoforms in the rat peripheral nervous system." *Neuroscience* **129**(3): 691-702.

Avraham, O., P. Y. Deng, S. Jones, R. Kuruvilla, C. F. Semenkovich, V. A. Klyachko and V. Cavalli (2020). "Satellite glial cells promote regenerative growth in sensory neurons." *Nat Commun* **11**(1): 4891.

Avraham, O., R. Feng, E. E. Ewan, J. Rustenhoven, G. Zhao and V. Cavalli (2021). "Profiling sensory neuron microenvironment after peripheral and central axon injury reveals key pathways for neural repair." *Elife* **10**.

Bosse, F., K. Hasenpusch-Theil, P. Küry and H. W. Müller (2006). "Gene expression profiling reveals that peripheral nerve regeneration is a consequence of both novel injury-dependent and reactivated developmental processes." *J Neurochem* **96**(5): 1441-1457.

Chen, L., Z. Wang, A. Ghosh-Roy, T. Hubert, D. Yan, S. O'Rourke, B. Bowerman, Z. Wu, Y. Jin and A. D. Chisholm (2011). "Axon regeneration pathways identified by systematic genetic screening in *C. elegans*." *Neuron* **71**(6): 1043-1057.

Ellman, D. G., T. J. Isaksen, M. C. Lund, S. Dursun, M. Wirenfeldt, L. H. Jørgensen, K. Lykke-Hartmann and K. L. Lambertsen (2017). "The loss-of-function disease-mutation G301R in the Na⁽⁺⁾/K⁽⁺⁾-ATPase α (2) isoform decreases lesion volume and improves functional outcome after acute spinal cord injury in mice." *BMC Neurosci* **18**(1): 66.

Fischer, D., V. Petkova, S. Thanos and L. I. Benowitz (2004). "Switching mature retinal ganglion cells to a robust growth state in vivo: gene expression and synergy with RhoA inactivation." *J Neurosci* **24**(40): 8726-8740.

Gill, K. P., S. S. Hung, A. Sharov, C. Y. Lo, K. Needham, G. E. Lidgerwood, S. Jackson, D. E. Crombie, B. A. Nayagam, A. L. Cook, A. W. Hewitt, A. Pébay and R. C. Wong (2016).

"Enriched retinal ganglion cells derived from human embryonic stem cells." *Sci Rep* **6**: 30552.

Hartl, D., A. R. Krebs, J. Jüttner, B. Roska and D. Schübeler (2017). "Cis-regulatory landscapes of four cell types of the retina." Nucleic Acids Res **45**(20): 11607-11621.

Jager, S. E., L. T. Pallesen, M. Richner, P. Harley, Z. Hore, S. McMahon, F. Denk and C. B. Vaegter (2020). "Changes in the transcriptional fingerprint of satellite glial cells following peripheral nerve injury." Glia **68**(7): 1375-1395.

Lindborg, J. A., N. M. Tran, D. M. Chenette, K. DeLuca, Y. Foli, R. Kannan, Y. Sekine, X. Wang, M. Wollan, I. J. Kim, J. R. Sanes and S. M. Strittmatter (2021). "Optic nerve regeneration screen identifies multiple genes restricting adult neural repair." Cell Rep **34**(9): 108777.

Lu, Y., Q. Shan, M. Ling, X. A. Ni, S. S. Mao, B. Yu and Q. Q. Cao (2022). "Identification of key genes involved in axon regeneration and Wallerian degeneration by weighted gene co-expression network analysis." Neural Regen Res **17**(4): 911-919.

Norsworthy, M. W., F. Bei, R. Kawaguchi, Q. Wang, N. M. Tran, Y. Li, B. Brommer, Y. Zhang, C. Wang, J. R. Sanes, G. Coppola and Z. He (2017). "Sox11 Expression Promotes Regeneration of Some Retinal Ganglion Cell Types but Kills Others." Neuron **94**(6): 1112-1120.e1114.

Pita-Thomas, W., T. M. Gonçalves, A. Kumar, G. Zhao and V. Cavalli (2021). "Genome-wide chromatin accessibility analyses provide a map for enhancing optic nerve regeneration." Sci Rep **11**(1): 14924.

Reverdatto, S., A. Prasad, J. L. Belrose, X. Zhang, M. A. Sammons, K. M. Gibbs and B. G. Szaro (2022). "Developmental and Injury-induced Changes in DNA Methylation in Regenerative versus Non-regenerative Regions of the Vertebrate Central Nervous System." BMC Genomics **23**(1): 2.

Tu, N. H., T. Katano, S. Matsumura, N. Funatsu, V. M. Pham, J. I. Fujisawa and S. Ito (2017). "Na(+)/K(+) -ATPase coupled to endothelin receptor type B stimulates peripheral nerve regeneration via lactate signalling." Eur J Neurosci **46**(5): 2096-2107.

Whitney, I. E., S. Butrus, M. A. Dyer, F. Rieke, J. R. Sanes and K. Shekhar (2022). "Vision-Dependent and -Independent Molecular Maturation of Mouse Retinal Ganglion Cells." Neuroscience.

Yang, M., L. Jian, W. Fan, X. Chen, H. Zou, Y. Huang, X. Chen, Y. G. Zhou and R. Yuan (2021). "Axon regeneration after optic nerve injury in rats can be improved via PirB knockdown in the retina." Cell Biosci **11**(1): 158.

Zibetti, C., S. Liu, J. Wan, J. Qian and S. Blackshaw (2019). "Epigenomic profiling of retinal progenitors reveals LHX2 is required for developmental regulation of open chromatin." Commun Biol **2**: 142.

REVIEWERS' COMMENTS:

Reviewer #1 (Remarks to the Author):

The authors provide in their rebuttal a more in depth explanation about the novelty of this manuscript.

I am also quite satisfied to Authors' reply and improvements to all my previous comments.

Reviewer #2 (Remarks to the Author):

The authors have addressed most of the queries except one the fonts and fonts sizes in main figures and supplementary figures are not consistent, some figures have smaller fonts and bold and some have regular fonts.